# Marine and terrestrial influences on ice nucleating particles during continuous springtime measurements in an Arctic oilfield location

Jessie M. Creamean[1,2], Rachel M. Kirpes[3], Kerri A. Pratt[3], Nicholas J. Spada[4], Maximilian Maahn[1,2], Gijs de Boer[1,2], Russell C. Schnell[5], Swarup China[6]

[1]Cooperative Institute for Research in Environmental Sciences, University of Colorado, Boulder, CO, 80309, USA
[2]Physical Sciences Division, National Oceanic and Atmospheric Administration, Boulder, CO, 80305, USA
[3]Department of Chemistry, University of Michigan, Ann Arbor, MI, 48109, USA
[4]Air Quality Research Center, University of California, Davis, CA, 95616, USA
[5]Global Monitoring Division, National Oceanic and Atmospheric Administration, Boulder, CO, 80305, USA
[6]Environmental and Molecular Sciences Laboratory, Pacific Northwest National Laboratory, Richland, WA, 99352, USA

*Correspondence to*: Jessie M. Creamean (jessie.creamean@noaa.gov)

**Abstract.** Aerosols that serve as ice nucleating particles (INPs) have the potential to modulate cloud microphysical properties and can therefore impact cloud radiative forcing and precipitation formation processes. In remote regions such as the Arctic, aerosol-cloud interactions are severely understudied yet may have significant implications for the surface energy budget and its impact on sea ice and snow surfaces. Further, uncertainties in model representations of heterogeneous ice nucleation are a significant hindrance to simulating Arctic mixed-phase cloud processes. We present results from a campaign called INPOP (Ice Nucleating Particles at Oliktok Point), which took place at a U.S. Department of Energy Atmospheric Radiation Measurement (DOE ARM) facility in the northern Alaskan Arctic. Three time- and size-resolved aerosol impactors were deployed from 1 Mar to 31 May 2017 for offline ice nucleation and chemical analyses and were co-located with routine measurements of aerosol number and size. The largest particles (i.e., $\geq 3$ μm or "coarse mode") were the most efficient INPs by inducing freezing at the warmest temperatures. During periods with snow- and ice-covered surfaces, coarse mode INP concentrations were very low (maximum of $6 \times 10^{-4}$ L$^{-1}$ at –15 ˚C), but higher concentrations of warm temperature INPs were observed during late May (maximum of $2 \times 10^{-2}$ L$^{-1}$ at –15 ˚C). These higher concentrations were attributed to air masses originating from over open Arctic Ocean water and tundra surfaces. To our knowledge, these results represent the first INP characterization measurements in an Arctic oilfield location and demonstrate strong influences from mineral and marine sources despite the relatively high springtime pollution levels. Ultimately, these results can be used to evaluate the anthropogenic and natural influences on aerosol composition and Arctic cloud properties.

## 1 Introduction

Aerosols are an important component of the atmospheric system through their various impacts on climate. Depending on the type (Satheesh and Moorthy, 2005), aerosols directly scatter and/or absorb radiation, thereby affecting the atmospheric energy budget (Boucher et al., 2013). Notably, the largest uncertainty of the energy budget (i.e., radiative forcing estimate) is the

aerosol indirect effect. As cloud condensation nuclei (CCN) and ice nucleating particles (INPs), aerosols influence atmospheric radiation through modulation of the microphysics of cloud droplets and ice crystals. Aerosol-induced microphysical modifications influence cloud lifetime and albedo, as well as the production of more or less precipitation (Albrecht, 1989; Twomey, 1977). However, constraining aerosol-cloud impacts in climate models, specifically when parameterizing INPs in mixed phase cloud (MPC) systems, remains a significant challenge due to limited observations (Cziczo et al., 2017; Coluzza et al., 2017; DeMott et al., 2010; Kanji et al., 2017; Korolev et al., 2017).

The efficacy of an aerosol to serve as an INP largely depends on its composition (i.e., chemical, mineral, or biological makeup), morphology, and size, and thus, its source. INPs nucleate ice through pathways dependent upon temperature, saturation with respect to ice, and the INP size and composition (Hoose and Möhler, 2012). Immersion freezing is the most relevant to MPC formation and requires that INPs initially serve as CCN (Hande and Hoose, 2017). Aerosols such as mineral dust, soil dust, sea salt, volcanic ash, black carbon from wildfires, and primary biological aerosol particles (PBAPs) have been shown to serve as INPs (Murray et al., 2012; Hoose and Möhler, 2012; Coluzza et al., 2017; Petters et al., 2009; DeMott et al., 1999; McCluskey et al., 2014; Conen et al., 2011). Among these, dust and PBAPs are the most adroit INPs found in the atmosphere (Murray et al., 2012; Coluzza et al., 2017). PBAPs originating from certain bacteria and vegetative detritus are the most efficient INPs known, capable of initiating freezing near −1 °C, while most PBAPs (e.g., pollen, fungal spores, algae, and diatoms) tend to nucleate ice at temperatures similar to those of mineral dust but warmer than sea salt or volcanic ash (Despres et al., 2012; Murray et al., 2012; Hoose and Möhler, 2012; Tobo et al., 2014; Hader et al., 2014b; O'Sullivan et al., 2014; Creamean et al., 2014; Creamean et al., 2013; Hill et al., 2016; Tesson et al., 2016; Alpert et al., 2011; Knopf et al., 2010; Umo et al., 2015; Durant et al., 2008; Fröhlich-Nowoisky et al., 2015; McCluskey et al., 2014). However, dusts can serve as atmospheric shuttles for small microbes, enabling these particle mixtures to behave more efficiently as INPs compared to the dust alone (Conen et al., 2011; Creamean et al., 2013; O'Sullivan et al., 2014). Consequently, PBAPs have the potential to play a crucial role in cloud ice formation (Creamean et al., 2014; Creamean et al., 2013; Pratt et al., 2009) and precipitation enhancement (Morris et al., 2004; Bergeron, 1935; Christner et al., 2008; Morris et al., 2014; Morris et al., 2017; Stopelli et al., 2014), particularly in the presence of supercooled water or large cloud droplets. PBAPs have been a key focus of recent ice nucleation studies, yet estimating their global impact remains a challenge due to: 1) a dearth of observations in time and space and 2) a poor understanding of sources, flux, and abundance of PBAPs in the Earth-atmosphere system (Hoose et al., 2010a).

Little is known about marine emissions of biological INPs—most studies of such aerosols to date have examined terrestrial sources (Burrows et al., 2013; Murray et al., 2012). Recent laboratory, field, and modelling studies have evaluated the mechanical emission processes, ice nucleation efficiency, and concentrations of INPs generated from marine environments (Knopf et al., 2010; Mason et al., 2016; McCluskey et al., 2017). Results from these studies imply that PBAPs are most relevant to ice formation at MPC temperatures (i.e., > −10 °C), whereas sea salt aerosols nucleate ice in both deposition and immersion

modes at temperatures relevant for cirrus clouds (i.e., < −38 °C) (Schill and Tolbert, 2014). Conclusions on the role of PBAPs in ice nucleation and precipitation are equivocally based on results from climate modelling, with some studies implying they are insignificant on a global scale (Hoose et al., 2010a; Sesartic et al., 2012; Hoose et al., 2010b), while others have found them to be potentially important (Phillips et al., 2009; Phillips et al., 2008; Vergara-Temprado et al., 2017). An inherent limitation of these pioneering studies is that results were validated by a handful of terrestrial-based observations of biological INPs. This, coupled with the fact that the Earth's surface is 70% covered by ocean, implies that studies using global climate simulations have ignored a potentially significant global source of INPs.

Previous work by Leck and Bigg in the High Arctic were among the first to elucidate the potentially paramount role of the ocean as a source of efficient INPs (Leck and Bigg, 2005; Bigg and Leck, 2001, 2008; Bigg, 1996), suggesting bacteria and fragments of marine organisms were responsible for the ice nucleating contribution to their samples. Schnell (1975) concluded that zones of profuse marine phytoplankton growth may release large numbers of INPs into the atmosphere based on measurements conducted on laboratory-cultured marine phytoplankton. However, new studies that provide information on key sources of INPs *and* their impacts on Arctic MPCs are necessary to assess the direct contribution from the marine environment. Irish et al. (2017) and Wilson et al. (2015) report enhanced ice nucleation activity of particulate matter in the surface microlayer and bulk seawater from the Arctic Ocean. Both studies concluded that the enhanced INP concentrations measured were attributed to heat-labile biological material and organic material associated with phytoplankton cell exudates, respectively. In addition, sources of INPs at Arctic coastal locations have been measured during the summer by Fountain and Ohtake (1985), Mason et al. (2016), and Conen et al. (2016) in Alaska, Canada, and Norway, respectively. Fountain and Ohtake (1985) did not comment on exact sources of INPs they measured at Utqiaġvik (formally Barrow), but determined INPs during episodic increases in concentration to be either from the Alaskan interior or long-range transported from Eurasia. Mason et al. (2016) did not comment on the sources of their size-resolved INPs at Alert between Mar and Jul but determined fewer local sources of aerosols as compared to their midlatitude coastal locations. Conen et al. (2016) attributed a fraction of their observed warm-temperature INPs (i.e., INP concentrations at –8 °C) from May to Sep were fungal spores and that most INPs were aerosolised locally by the impact of raindrops on plant, litter, and soil surfaces. Their measurement site was 219 meters above mean sea level (m a.m.s.l.) and 20 km from the actual coast, thus, could be the reason terrestrial sources were more influential at their site. To date, only a handful of studies have evaluated Arctic INPs in coastal or marine environments.

The Arctic is a remarkable region with regard to many atmospheric processes, but especially with regard to ice nucleation because: 1) the Arctic is mostly ocean, which may be the most prolific regional source of biological INPs, 2) interruption of ocean-atmospheric exchange by sea ice cover likely affects seasonally-dependent patterns in the emission of PBAPs/INPs, and 3) aerosol-cloud interactions could have implications important to understanding declining trends in sea ice and snow coverage. Overall, limited information on aerosol-cloud processes (Lubin and Vogelmann, 2006; Garrett and Zhao, 2006), contradictory modelling results with regard to the importance of biological INPs (Szyrmer and Zawadzki, 1997), and a critical need to

understand the role of aerosol particles in determining cloud phase (Bergeron, 1935; Wegener, 1911; Findeisen, 1938) motivate the need for additional observations to constrain the abundance and ice-nucleating properties of aerosols in the Arctic (Murray et al., 2012). Here, we present a 3-month record of continuous time- and size-resolved INP concentrations at Oliktok Point, a coastal site in Alaska situated within the North Slope of Alaska (Prudhoe Bay) oilfields, which have been shown to be strong and localized sources of pollutants. Recent studies have indicated that sources of aerosol in this region include carbonaceous combustion and aged sea spray aerosol measured during the winter and summer (Gunsch et al., 2017; Kirpes et al., 2018) and newly-formed particles from Prudhoe Bay gaseous emissions (Creamean et al., 2018a; Maahn et al., 2017; Kolesar et al., 2017), in which the anthropogenic pollutant aerosol has been linked to increased local cloud droplet concentrations over the Beaufort Sea (Hobbs and Rangno, 1998). Although anthropogenic aerosol may be the dominant type in this region, Gunsch et al. (2017) and Kirpes et al. (2018) also observed a significant fraction of supermicron aerosol to be fresh sea spray aerosol. Additional sources of aerosol observed at higher altitudes over Prudhoe Bay include regional wildfires and long-range transported pollution, although, these were measured during the summer (Creamean et al., 2018a). However, INPs and their sources have yet to be evaluated in this region. Here, we employ a comprehensive combination of size-resolved INP measurements, single-particle chemistry, bulk aerosol chemistry, local meteorology, regional scale transport, and sea ice and land cover conditions to assess INP sources in Prudhoe Bay. Unique observations of an increase in what were likely marine- and terrestrial-sourced INPs are discussed. This is the first time INP measurements have been conducted in an Arctic oilfield, and we demonstrate how efficient, mineral and marine-sourced INPs are likely important in such a relatively polluted Arctic location.

## 2 Methods

### 2.1 Study location and dates

INPOP was conducted at the Third Atmospheric Radiation Measurement (ARM) Mobile Facility (AMF-3) operated by the United States Department of Energy (DOE) in Oliktok Point, Alaska (70.51°N, 149.86°W, 2 m a.m.s.l.) (Creamean, 2017b). Oliktok Point is located in Prudhoe Bay (Figure 1), the third largest oilfield in North America (U.S. Energy Information Administration, 2015). The study time period was 1 Mar – 31 May 2017. Data from INPOP and the rest of the AMF-3 data record (2013 – present) are available on the ARM data archive (https://www.archive.arm.gov/discovery/) (Creamean, 2017a).

### 2.2 Sample collection

For offline ice nucleation analyses, daily aerosol samples were collected 11 Mar – 31 May 2017 (Table 1) using a 4-stage time- and size-resolved Davis Rotating-drum Universal-size-cut Monitoring (DRUM) single-jet impactor (DA400, DRUMAir, LLC.) (Cahill et al., 1987) housed in a 47 x 35.7 x 17.6 cm Pelican™ case (inset in Figure 1). The DRUM collected aerosol particles at four size ranges (0.15 – 0.34, 0.34 – 1.20, 1.20 – 2.96, and 2.96 – >12 μm in Stokes' equivalent diameter) and sampled at 26.7 L min$^{-1}$, equalling 38428 total L of air per day (i.e., per sample). Such size ranges cover a wide array of

aerosols, particularly those that serve as INPs (DeMott et al., 2010; Fridlind et al., 2012; Mason et al., 2016). Simultaneously, the large volume of air collected promotes collection of rarer warm temperature biological INPs, which may represent a lower fraction of overall INP concentrations (Mossop and Thorndike, 1966). Samples were deposited onto 20 x 190 mm strips of petrolatum-coated (100%, Vaseline®) Mylar™ (0.02 mm thick; DuPont®) substrate secured onto the rotating drums (20 mm thick, 60 mm in diameter) in each of the four stages at the rate of 7 mm per day (5 mm of sample streaked onto the Mylar followed by 2 mm of blank).

The DRUM was secured inside the AMF-3 container, with an approximately 7-m long sampling line (6.4 mm inner diameter static-dissipative polyurethane tubing; McMaster-Carr®) leading to outside of the container and connected to a plastic funnel inlet covered with loose mosquito netting to prevent rimed ice build-up or blowing snow from clogging the inlet. Aerosol collection was conducted at ambient relative humidity, which may affect particle size. The purpose of not drying the aerosol was to collect at sizes relevant to the particles in true environmental conditions. Daily rotation checks were conducted by disconnecting the DRUM from its pump and removing the stage caps to ensure the drums were rotating at the correct rate and sample substrates remained secured to each drum. Weekly checks of the inlet flow and pressure measured at each stage were conducted to ensure the orifices to each stage were not clogged and the DRUM pump was operating correctly.

The drums rotated for 24 to 26 days before sample substrates were changed (i.e., one full rotation of the drums), equalling three drum changes during INPOP. Sampled substrates were kept on the drums and stored frozen in a standard freezer until transport to Boulder, Colorado and stored frozen in a chest freezer for 5 to 7 months before offline INP analyses (see following section). This is the first time a DRUM has been used for offline INP analyses, but as we demonstrate, it is a useful sample collection method for long-term studies.

For offline single-particle analysis, daily samples were collected at 16.1 L min$^{-1}$ using a 3-stage DRUM impactor with particle size ranges 0.10 – 0.34 µm, 0.34 – 1.15 µm, and >1.15 µm on transmission electron microscopy (TEM) grids (Formvar carbon Type-B copper grids, Ted Pella, Inc.) attached to each of the three drums. The rotation was set at 3 mm per day, such that particles were deposited over the width of one TEM grid per 24-hour sampling period. Drums were changed once over the course of the INPOP study. Sampled substrates were kept on the drums and stored in the dark at ambient temperature until analysis (~3 months) (Laskina et al., 2015). Select daily samples corresponding to interesting case studies were analysed for discussion in the current paper. A third DRUM sampler collecting particles in 8 size bins (0.09 – 0.26 µm, 0.26 – 0.34 µm, 0.34 – 0.56 µm, 0.56 – 0.75 µm, 0.75 – 1.15 µm, 1.15 – 2.5 µm, 2.5 – 5.0 µm, and 5.0 – >12 µm) on Mylar™ for 12 hours per sample was co-located for bulk inorganic analysis (~2 months after collection). The flow rate was maintained at 9.6 L min$^{-1}$ with a drum rotation rate of 0.5 mm per day. Only one set of drums for this sampler was needed for the duration of INPOP. Samples were stored in the DRUM chambers during shipment and storage until analysis. The same sampling line set ups and routine checks as for the 4-stage DRUM were applied to the 3-stage and 8-stage DRUMs.

## 2.3 Drop freezing assays (DFA) for immersion mode INPs

In total, 216 daily samples were collected between the four different stages (herein, stages A, B, C, and D correspond to 2.96 – 12, 1.20 – 2.96, 0.34 – 1.20, and 0.15 – 0.34 µm particles, respectively. Due to contamination issues, stages B and D were not analysed from the first set of daily samples (see Table 1). Other missing data (i.e., samples not analysed) are due to occasional power outages at the AMF-3. Due to the large volume of samples, initially, one daily sample per week was analysed to assess a broader picture of INPs over the course of the spring months. Daily samples were analysed 22 – 29 May 2017 due to interesting aerosol events and source influences (as evidenced in the following sections) observed during this time period as described herein. Immediately prior to analysis, sample strips were cut into their daily segments and stored frozen in 29-mL sterile Whirlpak® bags for up to one week.

Immersion mode freezing was tested using a DFA cold plate apparatus. This cold plate technique was based on previous but slightly modified apparatuses (Hill et al., 2016; Tobo, 2016; Stopelli et al., 2014; Wright and Petters, 2013) and is discussed in detail by Creamean et al. (2018b). For brevity, we call this system the Drop Freezing Cold Plate (DFCP). Before analysing with the DFCP, 2 mL of ultrapure water (UPW; Barnstead™ Smart2Pure™ 6 UV/UF) were added to the sterile Whirlpak® bags containing sample segments to resuspend deposited particles. The bags were sealed and shaken at 500 rpm for two hours (Bowers et al., 2009). It is possible not all particles were removed during the extraction process, however, previous control testing indicates sufficient aerosol loading is resuspended (Creamean et al., 2018b). Yet, the fraction of material resuspended may depend on the overall loading. Copper discs (76 mm in diameter, 3.2 mm thick) were prepared by cleaning with isopropanol (99.5% ACS Grade, LabChem. Inc.), then coating with a thin layer of petrolatum (Tobo, 2016; Bowers et al., 2009). Following sample preparation, a sterile, single-use plastic syringe was used to draw 0.25 mL of the suspension and 100 drops were pipetted onto the petrolatum-coated copper disc, creating an array of ~2.5-µL aliquots. Drops were visually inspected for size; however, it is possible not all drops were the same exact volume, which could lead to a small level of indeterminable uncertainty (Alpert and Knopf, 2016). However, we have previously demonstrated that drop size variability does not impact freezing results (Creamean et al., 2018b). Previous studies have elucidated that drops need to be orders of magnitude different in volume to significantly perturb the freezing temperature from drop size, alone (Hader et al., 2014a; Bigg, 1953; Langham and Mason, 1958). The copper disc was then placed on a thermoelectric cold plate (Aldrich®) and covered with a transparent plastic dome. Small holes in the side of the dome and copper disc permitted placement of up to four temperature probes using an Omega™ thermometer/data logger (RDXL4SD; 0.1 °C resolution and accuracy of $\pm$ (0.4% + 1 °C) for the K sensor types used). During the test, the cold plate was cooled variably within a 1 – 10 °C min$^{-1}$ range from room temperature until around −30 °C. Control experiments with UPW at various cooling rates within this range show no discernible dependency of drop freezing on cooling rate (Creamean et al., 2018b), akin to previous works (Wright and Petters, 2013; Vali and Stansbury, 1966).

A +0.33 °C correction factor was added to any temperature herein and an uncertainty of 0.15 °C was added to the probe accuracy uncertainty based on DFCP characterization testing presented in Creamean et al. (2018b), to account for the temperature difference between the measurement (i.e., in the plate centre) and actual drop temperature. Frozen drops were detected visually, but the freezing temperature and cooling rate of each drop frozen was recorded through custom software.

5   The test continued until all 100 drops were frozen or when the system reached approximately −30 °C. Each sample was tested three times with 100 new drops for each test. From each test, the fraction frozen and percentage of detected frozen drops were calculated. For INPOP, 71 – 100% of the 100 drops were visually detected and recorded as frozen for each test. The results from the triplicate tests were then binned every 0.5 °C to produce one spectrum per sample. Cumulative INP spectra were calculated using the equation posed by Vali (1971) and adjusted to account for the total volume of air per sample:

$$[INPs(T)](L^{-1}) = \frac{\ln N_o - \ln N_u(T)}{V_{drop}} \times \frac{V_{suspension}}{V_{air}}$$

where $N_o$ is the total number of drops, $N_u(T)$ is the number of unfrozen drops at each temperature, $V_{drop}$ is the average volume of each drop, $V_{suspension}$ is the volume of the suspension (i.e., 0.75 mL total for the three tests), and $V_{air}$ is the volume of air per sample (38428 L) (Stopelli et al., 2014; Mason et al., 2016; Chen et al., 2018; Boose et al., 2016a).

**2.4 Single-particle and bulk aerosol composition**

15   Particles collected on TEM grids using the 3-stage DRUM were analysed for single-particle chemical composition and morphology using computer-controlled scanning electron microscopy with energy dispersive X-ray spectroscopy (CCSEM-EDX). Samples were analysed using a FEI Quanta scanning electron microscope with a field emission gun operating at 20 keV and equipped with a high angle annular dark field detector for particle size and morphology, and an energy dispersive X-ray X-ray spectrometer (EDAX, Inc.) and a Si(Li) detector (10 mm$^2$) for particle elemental composition of elements including 20   C, N, O, Na, Mg, Al, Si, P, S, Cl, K, Ca, Ti, V, Fe, Ni, and Zn. An average of 1000 particles was analysed per substrate. Data was analysed using k-means cluster analysis of the single-particle EDX spectra (Ault et al., 2012). Fifty clusters were combined into six classes of particle types, including fresh sea spray aerosol, aged sea spray aerosol, dust, organic aerosol, fly ash, and soot, based on similarity of elemental composition (Kirpes et al., 2018; Gunsch et al., 2017; Weinbruch et al., 2012).

Samples collected on the 8-stage DRUM were analysed for elemental concentrations using synchrotron-induced X-ray 25   fluorescence spectrometry (S-XRF) at the Lawrence Berkeley National Laboratory Advanced Light Source facility (VanCuren et al., 2012; Perry et al., 2004). The 1.5 GeV polarized X-ray beam provides very low background bremsstrahlung radiation and high sensitivity of elements including Na, Mg, Al, Si, P, S, Cl, Ar, K, Ca, Sc, Ti, V, Cr, Mn, Fe, Co, Ni, Cu, Zn, Ga, Ge, As, Se, Br, Rb, Sr, Y, Zr, Nb, Mo, and Pb. Spectra were acquired with a Vortex-60EX silicon drift detector (Hitachi High Technologies, USA) and were post-processed using both WinAXIL (Canberra, Belgium) and PyMca (Sole et al., 2007) 30   independently, for quality assurance. Further data reduction into timestamped concentrations was performed using custom

software (R Foundation for Statistical Computing, 2017). For specific comparisons, non-soil K was calculated as *non-soil-K = K − 0.45 × Fe* (Lewis et al., 1988) while soil K was calculated as *soil-K = K − non-soil-K*. These estimations ignore any contribution from biomass burning, which was a minor component of the observed single-particle designations.

## 2.5 Supporting observations and modelling

The AMF-3 at Oliktok Point includes the ARM Aerosol Observing System (AOS), which provides a wide range of aerosol and meteorological measurements (Jefferson, 2011). Data used here are publicly available in the ARM data archive and include aerosol number and size and wind direction and speed. Aerosol number concentrations were measured with a condensation particle counter (CPCf; 10 nm – 3 µm) and ultrafine CPC (CPCu; 3 nm – 3 µm) (TSI, Inc. models 3010 and 3025, respectively). Aerosol size distributions were measured with an ultrahigh-sensitivity aerosol sizer (UHSAS, Droplet Measurement Technologies, Inc.) for particles in the 60 nm – 1 µm size range. The AOS Surface Meteorology (AOSMET) data for atmospheric temperature, humidity, and wind speed and direction were measured via a Vaisala, Inc. WXT520 Weather Transmitter (Kyrouac, 2016). AOS data have a time resolution of 1 second. It is important to note that the aerosol from the AOS is dried to 40% relative humidity while the DRUM sampled aerosol at ambient relative humidity (RH), thus a possible small size disparity between the two inlets may exist. Daily snow and ice cover data were obtained from the National Snow & Ice Data Center. Snow cover data (http://nsidc.org/data/g02156#table3, Version 1) were derived from the Ice Mapping System (IMS) daily northern hemisphere snow analysis at a 1 km x 1 km resolution and are derived from a variety of data products including satellite imagery and in situ data (National Ice Center, 2008). Sea ice data (https://nsidc.org/data/g10005, Version 1) were derived from the Multisensor Analyzed Sea Ice Extent Advanced Microwave Scanning Radiometer 2 MASIE-AMSR2 (MASAM2) daily 4 km sea ice concentration product that is a blend of two other daily sea ice data products: ice coverage from the product at a 4 km grid cell size and ice concentration from the AMSR2 at a 10 km grid cell size (Fetterer et al., 2015). MASAM2 was used to meet a need for greater accuracy and higher resolution in ice concentration fields. Air mass backward trajectories (5-day) were calculated using the HYbrid Single Particle Lagrangian Integrated Trajectory model with the SplitR package for RStudio (https://github.com/rich-iannone/SplitR) (Draxler, 1999; Draxler and Rolph, 2011). Data from the Global Data Assimilation System (GDAS; 1° latitude-longitude; 3-hourly) from National Centers for Environmental Prediction (NCEP) were used as the meteorological fields in HYSPLIT simulations. Trajectories were initiated at 5 m above ground level (a.g.l.) (i.e., approximate height of the inlet) every 3 hours daily. Only hours within the sample day (i.e., starting at 16:00 UTC) were evaluated per each sample such that 18:00 and 21:00 on the same day and 00:00, 03:00, 06:00 09:00, 12:00, and 15:00 on the following day were considered per sample. Trajectories were only simulated for each day of May during sea ice melt off the Alaskan coast and during the 22 – 29 May case study.

## 3 Results and discussion

### 3.1 Atmospheric conditions during INPOP

Shifts in the aerosol population and atmospheric conditions throughout the course of INPOP were evident from the AOS measurements (Figure 2). Particle concentrations were high relative to what has been previously observed in the surrounding areas on the North Slope (e.g., Creamean et al., 2018a; Quinn et al., 2002), with the CPCu, CPCf and UHSAS measuring average concentrations of 2065, 1603, and 1483 cm$^{-3}$, respectively, over the entire study. These elevated concentrations are likely the result of the local industrial activities, which are prominent at Oliktok Point (Creamean et al., 2018a; Maahn et al., 2017; Gunsch et al., 2017). Akin to previous work on the North Slope during the Arctic haze, high particle concentrations were observed in April (e.g., Quinn et al., 2007), with averages of 2091, 3286, and 1526 cm$^{-3}$ for CPCu and 1702, 2736, and 907 cm$^{-3}$ for CPCf measured from March, April, and May, respectively. The monthly averaged UHSAS concentration was highest in March, at concentrations much higher than measured in May (3063, 1658, and 975 cm$^{-3}$ for March, April, and May, respectively). Mean particle size from the UHSAS was fairly consistent between the months (156.6±7.6, 149.9±9.9, and 143.6±7.3 nm, respectively). However, data from March may not be representative of the entire month due to a power outage in the first part of that month (i.e., missing data in Figure 2). As documented in previous studies (Law and Stohl, 2007; Stohl, 2006; Garrett et al., 2010; Di Pierro et al., 2013), changes in transport and precipitation patterns resulted in lower particle concentrations around late May. However, one interesting feature is the large variability in particle concentrations, particularly during late May, when hourly averaged concentrations ranged from 13 – 1486, 12 – 1111, and 35 – 1182 cm$^{-3}$ for the CPCu, CPCf, and UHSAS, respectively. Ambient temperature and relative humidity increased steadily over the course of the study and reached values indicative of coastal marine conditions towards the end of May (Cox et al., 2012).

### 3.2 Observation of a shift in springtime INP glaciation temperatures and concentrations in Prudhoe Bay

INP concentrations from the Arctic are typically very low compared to other regions (Kanji et al., 2017), and industrial regions are not thought to serve as prolific INP sources (e.g., Chen et al., 2018). Figure 3 shows the size-resolved INP concentrations from INPOP. A few key features are evident: (1) larger particles were more efficient INPs, especially at warmer temperatures, (2) larger particles exhibited a larger spread in INP spectra, and (3) late May samples contained particularly high concentrations of warm temperature (i.e., > −10 °C) INPs relative to March and April. INPOP INP concentrations were generally low, falling into the lower end of ranges reported by Kanji et al. (2017) and Petters and Wright (2015). Comparison of our results to previous ice nucleation studies in the Arctic are discussed in section 4.

The first feature discussed above has been reported by other size-resolved Arctic INP studies. Mason et al. (2016) report similar observations in Alert, Canada, where 95% and 70% of INPs were found at sizes > 1 μm and > 2.5 μm in diameter, respectively, and a median diameter of approximately 3 μm for all INPs at –15 ˚C. They also sampled at ambient RH, stating RH at 70% or greater can significantly reduce particle bounce via impaction sampling—RH during INPOP was typically > 70% (86±14%

on average, Figure 2) indicating the DRUM was minimally affected by particle bounce. In general, observational studies in locations outside of the Arctic also report a relationship between temperature and particle size, where supermicron or coarse mode aerosols are the most proficient INPs at warmer temperatures (Vali, 1966; references cited in Mason et al., 2016; Huffman et al., 2013; Conen et al., 2017). However, modelling studies suggest the mode of INPs can be as small as 500 nm

(DeMott et al., 2010; DeMott et al., 2015; Fridlind et al., 2012), while observational work suggests that nanometre-sized INPs are typically found attached to larger particles in the atmosphere (O'Sullivan et al., 2015). The second feature (larger spread in INP concentrations between samples with increasing size) indicates a larger variation in different aerosol sources or source strength (Seinfeld and Pandis, 2016; Mason et al., 2016) versus the smallest INPs observed which have similar spectra, indicating similar sources. Additionally, stage A of the DRUM has a wider size range (~9 µm) versus the smaller stages (~0.2

to 2 µm). The third reported feature was unexpected, given the characteristically polluted atmosphere in the Prudhoe Bay area. The 22 – 29 May period featured INP concentrations up to $4 \times 10^{-3}$ $L^{-1}$ at −10 °C and $2 \times 10^{-2}$ $L^{-1}$ at −15 °C, with warm temperature INPs apparent especially for the largest sizes (i.e., 2.96 – > 12 µm (stage A) and 1.21 – 2.96 µm (stage B)). The onset temperatures (up to −5 °C) were also much higher during this period for the two largest size cuts of the DRUM (Figure 4), indicating a presence of biological INPs (Murray et al., 2012; Kanji et al., 2017). The presence of biological INPs in Prudhoe

Bay in the spring are somewhat unexpected given the predominantly frozen surfaces. Despite this, the sizes of the observed INPs indicates a more local or regional origin due to reduced atmospheric lifetime (Jaenicke, 1980). The smallest sizes (0.34 – 1.21 µm (stage C) and 0.15 – 0.34 µm (stage D)) did not portray the same trend during late May. There, a higher fraction of particles was active at much colder temperatures (i.e., $INP_{-20}$ and $INP_{-25}$) as compared to $INP_{-10}$ or $INP_{-15}$, where temperatures are generally relevant to less efficient biological INPs (i.e., spores) and mineral dust (Murray et al., 2012; Haga et al., 2014;

Boose et al., 2016b). A previous study by Stone et al. (2007) demonstrated transport of Asian dust to Utqiaġvik during the spring, thus, introducing sources of colder temperature INPs even though the surface is predominantly frozen.

### 3.3 Open Arctic waters and snow melt over tundra as sources of observed transition in INP properties

Focusing on the month of May, it is clear that there are higher concentrations of warm temperature INPs, particularly for DRUM stages A and B (Figure 4). Stages C and D include, or are completely composed of, particles that are thought to be less

efficient INPs relative to particles with larger diameters (DeMott et al., 2010; DeMott et al., 2015; Fridlind et al., 2012), which is consistent with the low INP activity from such particles during INPOP. Stage A was especially unique, with the range of freezing temperatures becoming smaller over time. Snow coverage decreased sharply within 700 km of Oliktok Point on 8 May, while sea ice within 700 km dropped in fraction after 13 May (Figure 5e), which roughly aligns with the increase in onset freezing temperature for the largest particles in stage A and generally higher onset freezing temperatures for stages B and C.

The observed increase of warm temperature INPs at Oliktok Point can be attributed to air mass transport pathways and timescales over specific sources. Transport of regional air was quite slow and remained predominantly near the surface, as evidenced by the 5-day air mass back trajectories shown in Figure 6. At the beginning of May, frozen surfaces were prominent

across the entire region, including the areas over which air transported to Oliktok Point travelled. On 9 May, the marginal ice zone (MIZ) started to develop off the coast of Oliktok Point as evidenced by the < 100% sea ice coverage values in the 4-km data (i.e., areas of exposed open water indicated by the light purple shading in Figure 6b inset). The MIZ may be a combination of ice floes and/or open leads, but we cannot differentiate these features using the 4-km data. Onset temperatures for stages A and B started to increase on 9 May, coincident with the MIZ starting to expose open water. Starting 16 May, polynyas were observed west of the Canadian Archipelago and northeast of Oliktok Point, about 700 km away. Air mass trajectories show low-altitude transport over this region, corresponding to a slight increase in warm temperature INPs on that day and the warmest onset temperatures (Figure 5a). Due to the distance between Oliktok and the open water, particles within the size range of stage A likely gravitationally settled during the 5-day transit, yet such sources may still have provided small contributions of aerosol to the observed enhanced INPs as evidenced by the warmest onset temperatures and only slightly elevated INP concentrations as compared to samples collected before 16 May. Jaenicke (1980) showed that particles within the size range of stage A have typical atmospheric lifetimes on the order of minutes to days, particularly when in the boundary layer and near the surface. Based on the minimum and maximum ambient air temperatures observed at Oliktok Point on 16 May (–3 and 2 ˚C, respectively) and altitudes of the trajectories over the polynya northeast of Oliktok Point (40 – 90 hours back in time), basic terminal settling velocity calculations using on the Navier-Stokes equation indicate particles within the stage A range would have gravitationally settled within 11 seconds to 6 hours, indicating the polynyas were not the dominant source and the MIZ closer to Oliktok Point was more likely the source of the larger INPs. However, this simple calculation assumes starting temperature and does not consider vertical updrafts and downdrafts that may affect particle lifetime. On 22 May, there was a significant shift in air mass origin: air reaching Oliktok Point travelled only over the MIZ and not the polynyas to the northeast (Figure 6d inset). This shift corresponded to the highest INP concentrations observed from any sample analysed. Sources were generally regional in nature, with locations directly north of Oliktok Point until 26 May, after which they transitioned to originating predominantly south of Oliktok Point, from exposed tundra over the terrestrial North Slope and interior Alaska. Ambient air temperatures at Oliktok Point starting 26 May were relatively warm and fluctuated until 29 May (Figure 2), indicating air originating from over warmer surfaces (i.e., darker land versus brighter ice). Onset temperatures decreased following 26 May and INP concentrations remained high, but for colder freezing temperatures (i.e., –25 ˚C; Figure 4a). On 29 May, INP concentrations dropped but were still higher than Mar – early May samples, and air masses transitioned to originating from the northeast. Like 16 May, terminal settling velocity calculations indicate particles on this day were likely from closer to Oliktok Point, ranging from 13 seconds to 7 hours.

Recent work by May et al. (2016) has demonstrated that production of sea salt aerosol in the Arctic can occur under non-stagnant conditions from open leads (i.e., winds speeds > 4 m s$^{-1}$). The main mechanisms behind aerosol production from open ocean surfaces is bubble bursting from wind-induced wave breaking, although this process is far less studied over leads. A recent study by Gabric et al. (2018) describes generation of marine biogenic aerosol (MBA) from sea ice leads and the MIZ. Thus, other primary aerosols—such as bacteria or marine organisms that have been shown to serve as efficient INPs—may be

generated by the same mechanisms that produce sea salt aerosol and MBA. Recent studies by Wilson et al. (2015) and Irish et al. (2017) have shown that the Arctic Ocean surface microlayer and bulk seawater can harbor large concentrations of INPs, indicating physical mechanisms that generate aerosol from the surface waters may eject these INPs into the atmosphere. Additionally, several previous high Arctic ice nucleation studies have demonstrated that leads and other open water sources as vital to influencing atmospheric INP concentrations (Bigg, 1996; Bigg and Leck, 2001, 2008; Leck and Bigg, 2005). Based on our source analysis and a combination of conclusions from these previous studies, we conclude that INPs from open water features are likely produced via bubble bursting and are likely composed of bacteria or fragments of marine organisms.

Collocated single-particle and bulk chemistry measurements support the sources of the air masses during the transition period in late May (Figure 7). Here, we briefly discuss results from the compositional analyses in support of the ice nucleation measurements. Analysis of May periods by CCSEM-EDX identified marine, terrestrial, and combustion influences based on particle composition and air mass trajectories (Figure 7a – d). Particle types from marine (fresh and aged sea spray aerosol (SSA)) and terrestrial (dust) sources comprised the greatest number fractions of supermicron (> 1.15 µm) aerosol particles for all samples (54 % and 37 %, respectively). Each May sample demonstrated a large fraction of marine influence, with nearly 50% of supermicron particles, by number, comprised of fresh and aged SSA. The largest marine influence was observed on 23 May when air masses originated from the north over open water, local wind speeds were low and northerly, and INP concentrations, particularly in the warm temperature regime, were elevated compared to samples prior to 22 May. However, periods with more terrestrial influence were observed, based on increased number fractions of supermicron dust and southerly winds on 24 May (Figure 7e). Additionally, some periods experienced more combustion influence, from local industrial emissions, characterized by relatively greater fractions of organic aerosol, fly ash, and soot (Gunsch et al., 2017; Kirpes et al., 2018) observed by CCSEM-EDX on 28 May. The S-XRF data (0.75 – 5.0 µm) also demonstrated influences from marine and terrestrial sources, but had the highest marine signature (i.e., from fresh sea salt indicated by the Cl mass concentrations) 16 – 20 May when local winds were elevated (~ 10 m s$^{-1}$) and originated from the north over the Beaufort Sea. The S-XRF indicated the highest dust influence was observed from 25 – 30 May when episodic southerly winds from over the tundra influenced Oliktok Point (Figure 7e). The observed dust could be due to local road dust in addition to terrestrial dust sources along the air mass back trajectories.

Combined with air mass trajectory analysis, the single-particle and bulk chemistry support the conclusion that regional mineral and marine sources largely contributed to the enhanced warm temperature INP concentrations during late May. Very little contributes from soot and fly ash were observed (4% and 16% on 23-May and 28-May, respectively), indicating local pollution was not a large source of the particles in general. Additionally, soot and fly ash can serve as INPs, but at temperatures much lower than –15°C (Kanji et al., 2017; Murray et al., 2012). Aside from the composition, we would not expect pollution sourced from Prudhoe Bay (which tends to be sub-100 nm) to overlap with the sizes of the INPs observed (i.e., > 2.96 µm) (Creamean et al., 2018a; Maahn et al., 2017). Another possible source that has been shown to influence this region in the spring is long-

range transported aerosol. Recent work by Kylling et al. (2018) demonstrates surface dust impacts from several midlatitude sources to the entire Arctic is predominant during May. Yet, they also show that the major source of dust at the surface is from North American north of 60 ˚N, indicating regional source influences. Specific to our region, supermicron mineral dust has been shown to be transported during Arctic Haze as demonstrated by Quinn et al. (2002) at Utqiaġvik, but at very low mass concentrations relative to other aerosol species. Additionally, given the size in which we observed the INPs (> 2.96 μm), it is unlikely such large particles were transported from very distant sources, especially considering the air mass transport pathways were near the surface in the boundary layer for several days prior to arrival at the site. Thus, based on the combination of previous parallel studies, freezing temperatures, size, single-particle composition, bulk composition, local meteorology, and air mass transport, we demonstrate that it is unlikely local pollution largely influenced the INP concentrations during late May.

## 4 Comparison to other Arctic INP measurements

Immersion mode INP measurements in the Arctic are rare, but here we provide a synopsis of previous measurements in comparison to ours. Table 2 shows ground-based, shipborne, and airborne studies dating back to 1976 in locations throughout the Arctic during various seasons. We excluded a review of diffusional chamber INP measurements as these instruments predominantly measure deposition mode INPs and are thus not directly comparable to our results (Kanji et al., 2017; Cziczo et al., 2017). While the objective of this comparison is to determine whether our measurements align with previous immersion mode INP observations, a detailed review of Arctic INPs is outside the scope of this manuscript.

Overall, our concentrations at selected temperatures were within range of with those previously reported (Bigg, 1996; Borys, 1989; Conen et al., 2016; Mason et al., 2016; Fountain and Ohtake, 1985; Radke et al., 1976; Prenni et al., 2009), even though there are: (1) likely dependencies on time of year and location and (2) a spread by several orders of magnitude is apparent. Figure 8 shows the range of our results in comparison with those in Table 2. In general, our INP concentrations agree with or are on the lower end of those previously reported in the spring. One explanation could be that due to the proximity to the Prudhoe Bay oilfield, we are sampling in a location that is more polluted at the ground than those previously used to document Arctic haze (Borys, 1989; Mason et al., 2016; Radke et al., 1976). Additionally, Borys (1989) and Radke et al. (1976) measured via airborne platforms, in which they predominantly measured long-range transported Arctic Haze as compared to Arctic boundary layer aerosol. Other sources of variation in the results could be the result of discrepancies in the sample volumes and drop sizes used under different DFA techniques (Vali, 1971; Mossop and Thorndike, 1966). Also, unlike some of the studies presented in Table 2 and Figure 8, we collected substantial volumes of air per sample to enable measurement of the rare warm temperature INPs (38428 L in the present study versus 250 – 32400 L). Despite these differences, our concentrations are consistent with Bigg (1996), Conen et al. (2016), and DeMott et al. (2016), with those studies conducted during summer or fall. Those previous studies were apportioned to either marine or terrestrial sources, resulting in an analogous set of studies to our current effort. Overall, our measured INP concentrations are comparable to those previously conducted in a similar fashion

in the Arctic and predominantly fall within the overall global picture of INPs presented by Kanji et al. and (2017) and Petters and Wright (2015).

## 5 Summary and broader implications

We present the first INP measurements in an Arctic oilfield location and demonstrate how local and regional transport from
marine and terrestrial sources to an industrial region can introduce high concentrations of coarse, warm temperature INPs that are possibly of biological origin. Three time- and size-resolved aerosol impactors were deployed from 1 Mar to 31 May 2017 for offline ice nucleation and chemical analyses and were co-located with routine measurements of aerosol number, size, chemistry, and radiative properties. The largest particles (i.e., $\geq 3$ μm or "coarse mode") were the most efficient INPs. During periods with snow- and ice-covered surfaces, coarse mode INP concentrations were very low (maximum of $6 \times 10^{-4}$ L$^{-1}$ at –15
˚C), but higher concentrations of warm temperature INPs were observed during late May (maximum of $2 \times 10^{-2}$ L$^{-1}$ at –15 ˚C). These higher concentrations are attributed to air masses originating from over open Arctic Ocean water and tundra surfaces, and were likely primary marine aerosol and dust, respectively. In general, our concentrations agree with similar immersion mode ice nucleation evaluations at other Arctic locations, even though such studies vary in terms of location, volume of air per sample, drop freezing technique, and time of year.

Although our measurements were conducted at the ground level, such particles could influence Arctic MPC (AMPC) formation under conditions where the cloud is coupled to the surface by a well-mixed boundary layer. Previous long-term studies have evaluated the annual cycle of AMPC properties and, relevant for our measurements, have concluded that climatologically a large number of AMPC occur over the North Slope of Alaska in May (up to 84 – 90% cloud fraction), with cloud bases over the North Slope reaching down below 500 – 700 m (Shupe, 2011; Shupe et al., 2010; Dong et al., 2010). Specifically, AMPC
fraction is at a maximum in the fall, followed by the spring, with the lowest cloud bases occurring over the Arctic Ocean (Shupe et al., 2006; Shupe et al., 2005; Shupe, 2011). Combined, these studies indicate that INPs from surface sources may be important for such low, persistent clouds. Further, average cloud temperatures in the Arctic have been shown to be relatively warm in the spring and into the summer (up to –10 ˚C on average) (de Boer et al., 2009; Shupe, 2011), indicating that biological INPs from Arctic sources are relevant to AMPC conditions typical for this time of year.

We have demonstrated that the spring to summer transition on the North Slope and beyond can have implications for a shift in INP properties, potentially impacting cloud phase, precipitation amounts, and cloud lifetime. Given significant shifts in broadband radiation regimes starting March manifesting in a sign change in cloud radiative forcing (CRF) from winter and spring months to summer (Dong et al., 2010), understanding aerosol-cloud interactions is crucial during this time of year. Additionally, with increasing likelihood of earlier melting of frozen surfaces in the Arctic in coming decades (Markus et al.,
2009; Stroeve et al., 2014; Screen and Simmonds, 2010), the influence of INP on CRF may be shifting in season along with the sign of CRF, with the latter resulting from a complex mix of shifting atmospheric temperature and surface albedos.

Understanding the role of INPs in modulating CRF is important as this forcing can act as a positive or negative feedback on the acceleration of surface melt, and the timing of such melt can potentially lead to impacts on a variety of features, midlatitude weather and climate (Cohen et al., 2018). Together, these items demonstrate a strong need to continue to improve understanding of high latitude INPs, with additional INP concentration and characterization measurements required to confirm the findings presented here and evaluate this cycle beyond spring months.

## Acknowledgements

We acknowledge Fred Helsel, Bruce Edwardson, and Mark Ivey for assistance in study planning, logistics, and setup. Additionally, we thank AMF-3 operators David Oaks, Ben Bishop, Joshua Remitz, and Wessley King for ensuring successful operation of the DRUM during INPOP. We also thank the members of the Oliktok Point Site Science team for their valuable feedback during data analysis and acknowledge the NOAA Air Resources Laboratory (www.arl.noaa.gov/ready.php) and Rich Iaonnone for the provision of the SplitR HYSPLIT transport model (https://github.com/rich-iannone/SplitR). Andrew Ault (University of Michigan) is acknowledged for discussions of aerosol sampling for INPOP. Funding was provided by US DOE ARM (2016-6841, 2016-6884, and 2016-6875) and Atmospheric Systems Research (ASR) (DE-SC0013306) programs for sample collection/analysis and data interpretation, respectively. Additionally, funding was provided by the NOAA Physical Sciences Division (PSD) to support data and sample analyses. K.A. Pratt and R. Kirpes were supported, in part, by an Early-Career Research Fellowship from the Gulf Research Program of the National Academies of Sciences, Engineering, and Medicine. The content is solely the responsibility of the authors and does not necessarily represent the official views of the Gulf Research Program of the National Academies of Sciences, Engineering, and Medicine. CCSEM-EDX analyses were performed at the Environmental Molecular Sciences Laboratory (EMSL), a national scientific user facility located at the Pacific Northwest National Laboratory (PNNL) and sponsored by the Office of Biological and Environmental Research of the US Department of Energy (DOE). PNNL is operated for DOE by Battelle Memorial Institute under contract no. DE-AC06-76RL0 1830. Travel funds to PNNL were provided by the Michigan Space Grant Consortium

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

**Table 1: Dates and start times of sample collection during INPOP (2017) for samples analysed for INPs. Samples correspond to 1 full day (24 hours from the start time). The "INPs" column corresponds to the stages that were analysed via the DFCP per daily sample. A = 2.96 – >12 µm, B = 1.21 – 2.96 µm, C = 0.34 – 1.21 µm, and D = 0.15 – 0.34 µm.**

| Date | Start (UTC) | INPs |
|---|---|---|
| 11-Mar | 21:00 | A, C |
| 18-Mar | 20:00 | A, C |
| 25-Mar | 20:00 | A, C |
| 01-Apr | 20:00 | A, C |
| 17-Apr | 16:00 | A, B, C, D |
| 24-Apr | 16:00 | A, B, C, D |
| 02-May | 16:00 | A, B, C, D |
| 09-May | 16:00 | A, B, C, D |
| 16-May | 16:30 | A, B, C, D |
| 22-May | 16:30 | A, B, C, D |
| 23-May | 16:30 | A, B, C, D |
| 24-May | 16:30 | A, B, C, D |
| 25-May | 16:30 | A, B, C, D |
| 26-May | 16:30 | A, B, C, D |
| 27-May | 16:30 | A, B, C, D |
| 28-May | 16:30 | A, B, C, D |
| 29-May | 16:30 | A, B, C, D |

**Table 2: Comparison of atmospheric immersion mode INPs measurements presented here with previous studies in the Arctic. Study reference, location, dates, and average INP concentrations converted to $L^{-1}$ reported for up to the four different temperatures shown. The superscripts g, s, and a represent ground-based, shipborne, and airborne measurements, respectively. "None" refers to no reported concentrations (or observed, in the case of the current study) measured at that temperature.**

| Study | Location | Dates | Max volume of air (L) | INPs$_{-10}$ ($L^{-1}$) | INPs$_{-15}$ ($L^{-1}$) | INPs$_{-20}$ ($L^{-1}$) | INPs$_{-25}$ ($L^{-1}$) |
|---|---|---|---|---|---|---|---|
| Bigg (1996) | High Arctic[g] | early Aug 1991 | 3000 | none | $1 \times 10^{-2}$ | none | none |
| | | Aug – Oct 1991 | | none | $3 \times 10^{-3}$ | none | none |
| Borys (1989) | Alaska, Canada, Greenland[a] | Apr 1986 | 1400 | none | $2 \times 10^{-2}$ | none | $5 \times 10^{-1}$ |
| Conen et al. (2016) | Norway[g] | Jul 2015 | 24000 | $6 \times 10^{-4}$ | $7 \times 10^{-3}$ | none | none |
| DeMott et al. (2016) | Bering Sea[s] | Summer 2012 | 13500 | none | $3 \times 10^{-3}$ | $3 \times 10^{-2}$ | none |
| Fountain and Ohtake (1985) | Alaska[g] | Aug 1978 – Apr 1979 | 250 | none | none | $1 \times 10^{-1}$ | none |
| Mason et al. (2016) | Canada[g] | Mar – Jul 2014 | 32400[*] | none | $5 \times 10^{-2}$ | $2 \times 10^{-1}$ | 1 |
| Prenni et al. (2009) | Alaska[a] | Oct 2004 | na[**] | $2 \times 10^{-1}$ | none | $4 \times 10^{-1}$ | none |
| Radke et al. (1976) | Alaska[a] | Mar 1970 | 3000 | none | none | $2 \times 10^{-2}$ | none |
| This study (2018) | Alaska[g] | Mar – May 2017 | 38428 | $8 \times 10^{-4}$ | $5 \times 10^{-3}$ | $2 \times 10^{-2}$ | $4 \times 10^{-2}$ |
| | | Mar 2017 | | none | $7 \times 10^{-4}$ | $6 \times 10^{-3}$ | $3 \times 10^{-2}$ |
| | | Apr 2017 | | $7 \times 10^{-5}$ | $2 \times 10^{-3}$ | $1 \times 10^{-2}$ | $7 \times 10^{-2}$ |
| | | May 2017 | | $1 \times 10^{-3}$ | $8 \times 10^{-3}$ | $2 \times 10^{-2}$ | $4 \times 10^{-2}$ |
| | | late May 2017 | | $2 \times 10^{-3}$ | $1 \times 10^{-2}$ | $3 \times 10^{-2}$ | $4 \times 10^{-2}$ |

[*]Not directly provided by citation. Estimated from average total volume of air per sample and standard flow rate specification for the sampler.

[**]Immersion freezing assumed to be dominant mode of INP concentrations measured via an online continuous flow diffusion chamber.

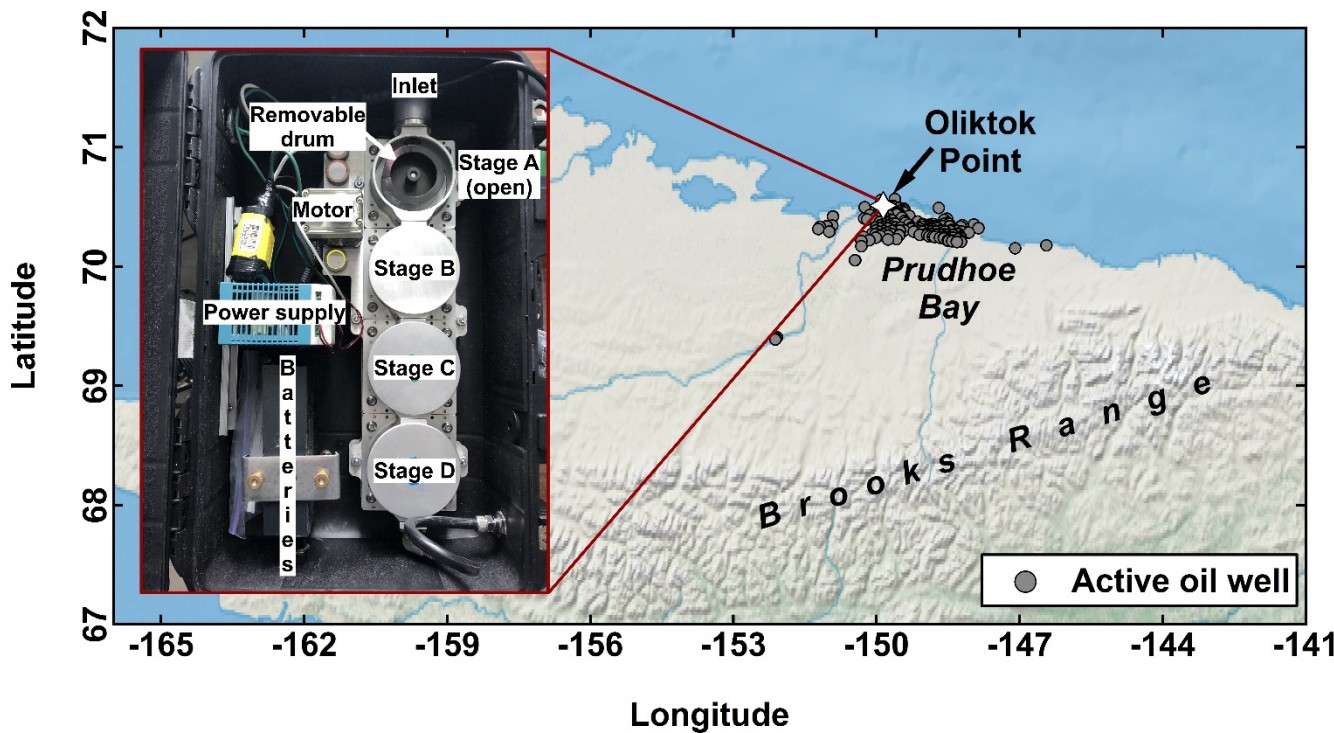

**Figure 1: Map of the North Slope of Alaska highlighting Oliktok Point and oil wells that are active in Prudhoe Bay (data obtained from http://doa.alaska.gov/ogc/publicdb.html in Mar 2017). The approximate areas of Prudhoe Bay and the Brooks Mountain range are shown. Inset shows the inside of the DRUM case, with stage A exposed (i.e., cover removed) and major components labelled.**

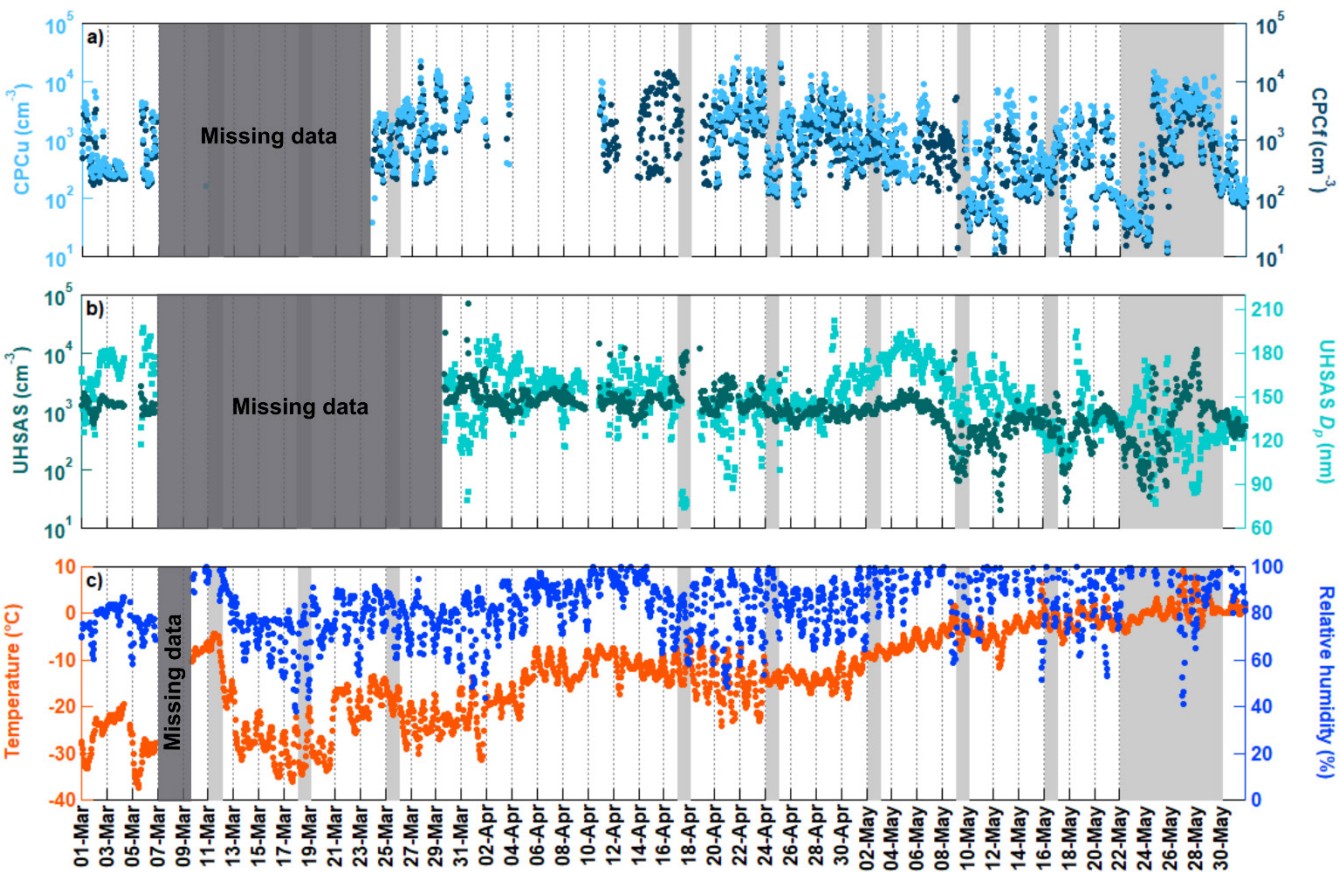

**Figure 2: Time series of hourly-averaged aerosol number and meteorological variables from AOSMET during 1 Mar – 31 May 2017, including aerosol number concentrations measured by a) the condensation particle counter (CPCf) and ultrafine CPC (CPCu) and b) the ultrahigh-sensitivity aerosol sizer (UHSAS). Average hourly mean particle diameter ($D_p$) is also shown from the UHSAS in b). Meteorological data include c) relative humidity and air temperature. Light grey shading represents days where samples were analysed for INP concentrations. Dark grey shading represents times with missing data.**

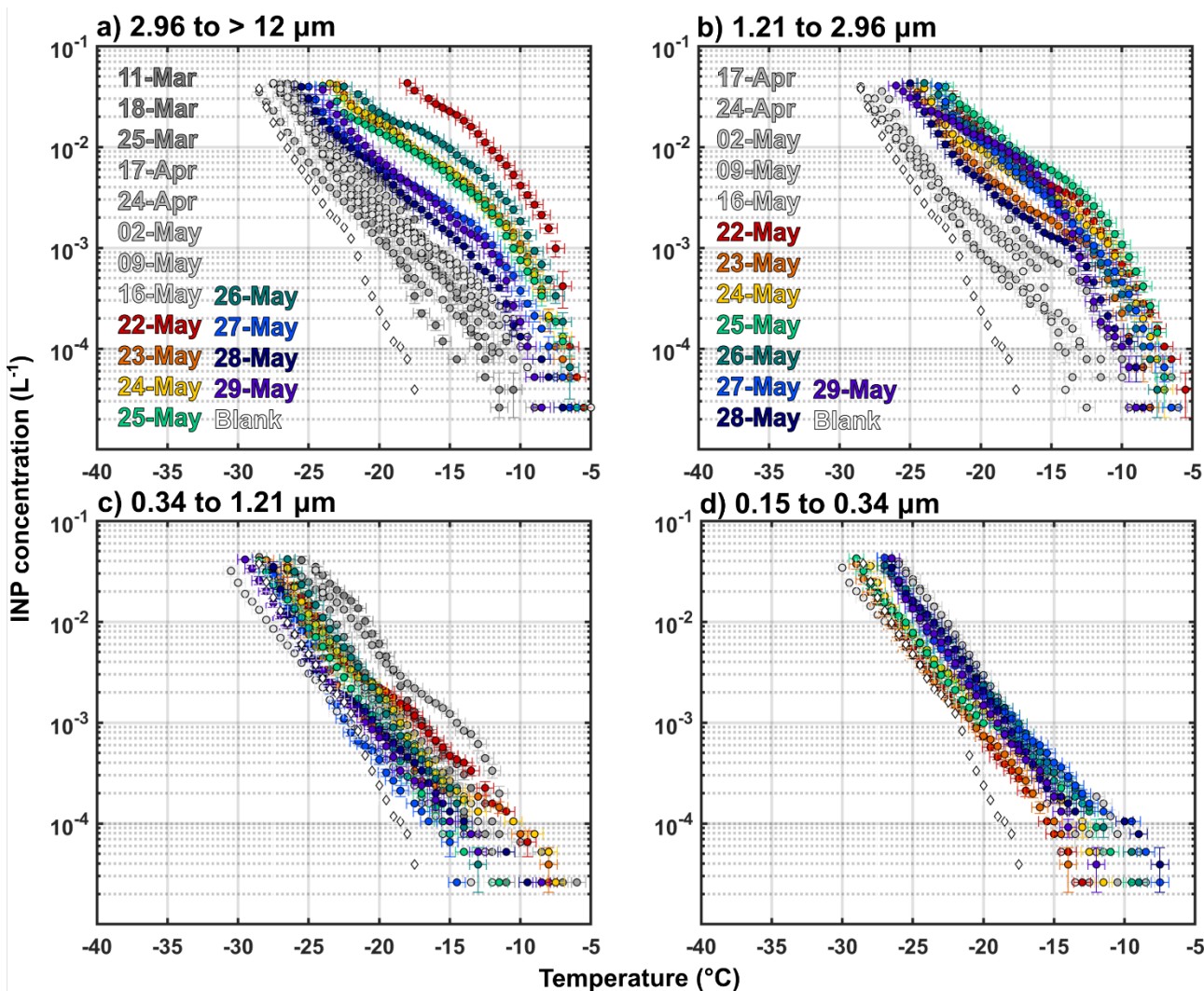

Figure 3: Cumulative INP spectra for the four size-ranged samples of the DRUM from INPOP (panels a – d represent stages A – D, respectively). One daily sample per week is shown from 11 Mar to 16 May and 17 Apr to 16 May for stages A/C and B/D, respectively. Daily samples are shown from 22 May – 29 May for all stages. White diamond markers denote the blank Mylar sample in UPW prepared in the same manner as the samples. This blank spectrum applies to all size ranges.

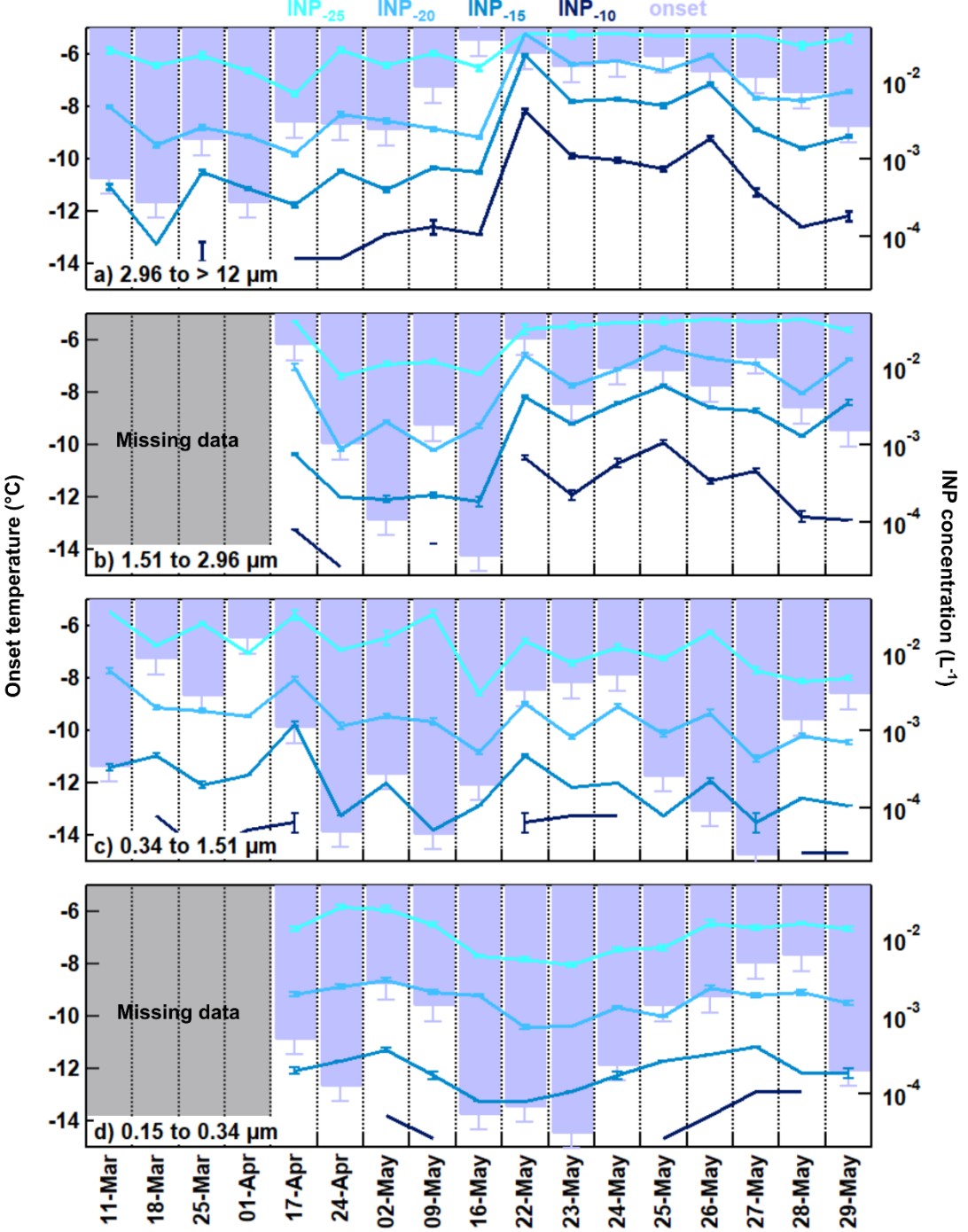

**Figure 4: Onset freezing temperatures and INP concentrations at −10 °C, −15 °C, −20 °C, and −25 °C for DRUM stages a) A, b) B, c) C, and d) D for each sample collected during INPOP. Dark grey shading represents missing data. Error bars for the onset temperature represent the probe uncertainty and plate correction and standard deviation for the INP concentrations.**

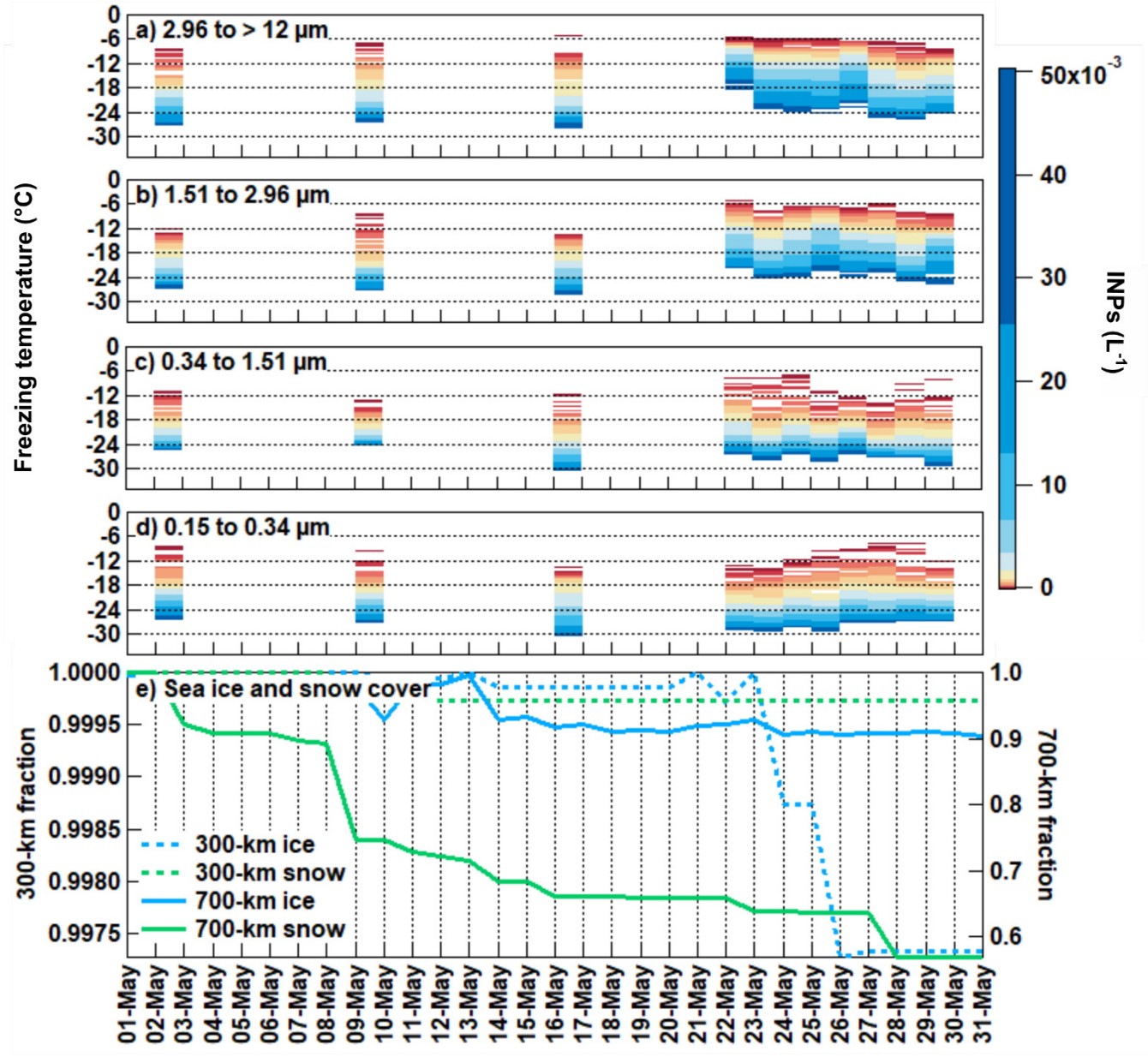

**Figure 5: Time series of INP freezing temperatures coloured by INP concentrations (coloured bands, left axis) for DRUM stages a) A, b) B, c) C, and d) D. Note that the INP concentrations are in log scale. Also shown are the e) sea ice and snow coverage within a 300-km and 700-km radius from Oliktok Point. Data for the month of May are shown (http://nsidc.org/data/g02156#table3).**

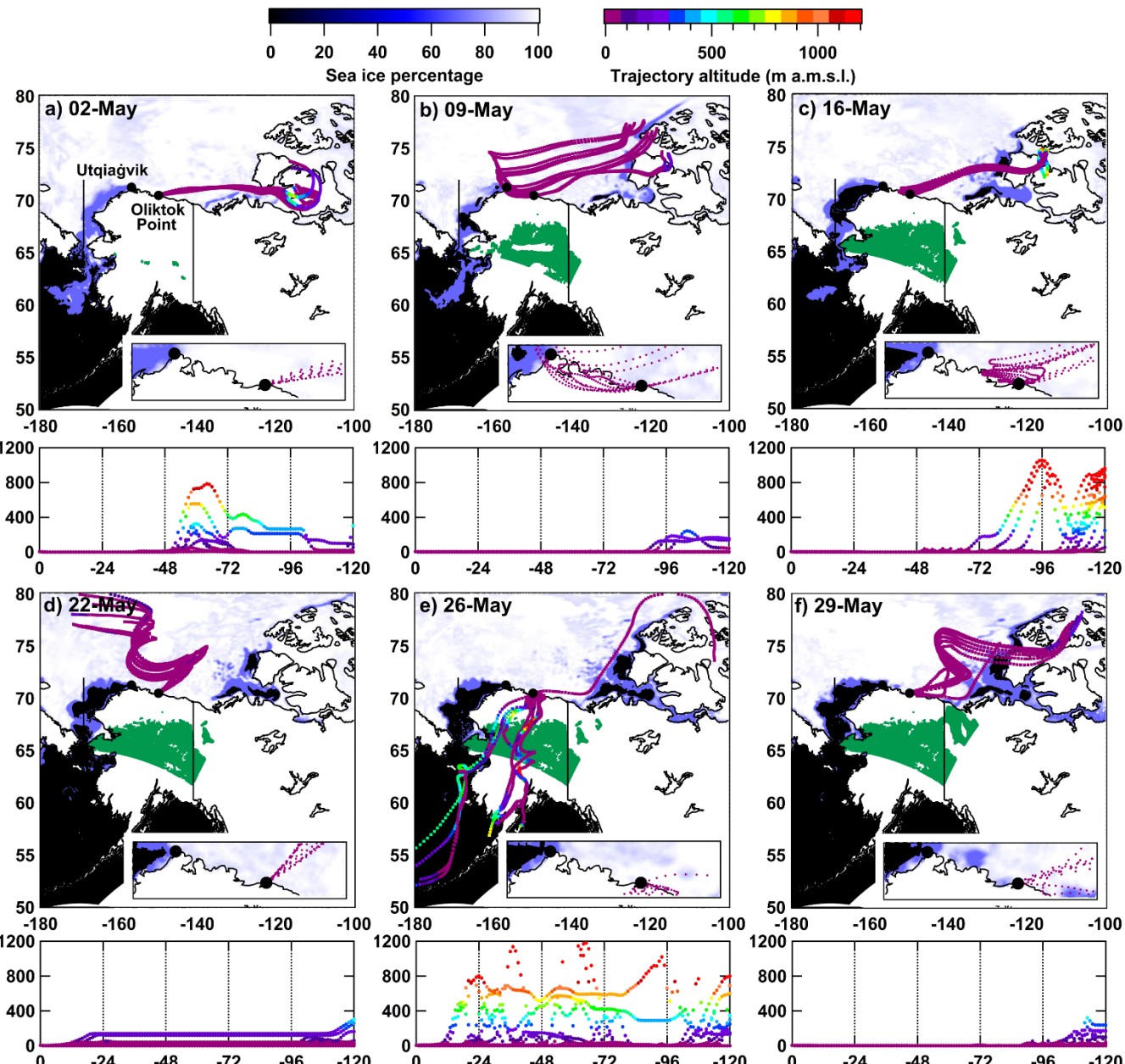

**Figure 6:** 5-day air mass backward trajectories initiated every 3 hours and land cover (e.g., sea ice, snow, ocean, and open land) for some of the case days in May with INP data. Each date contains the map and corresponding time-height cross section of the 5-day trajectories (shown below each map). The x- and y- axes for the maps represent degrees longitude and latitude, respectively, while those axes for the time-height cross sections represent hours back in time and altitude (m a.m.s.l.), respectively. The locations of Utqiaġvik and Oliktok Point are labelled in panel a) but indicated in all panels by the black circles. On land, white indicates snow covered surfaces while green indicates open land (e.g., tundra). Land data are missing in southern Alaska and most of Canada.

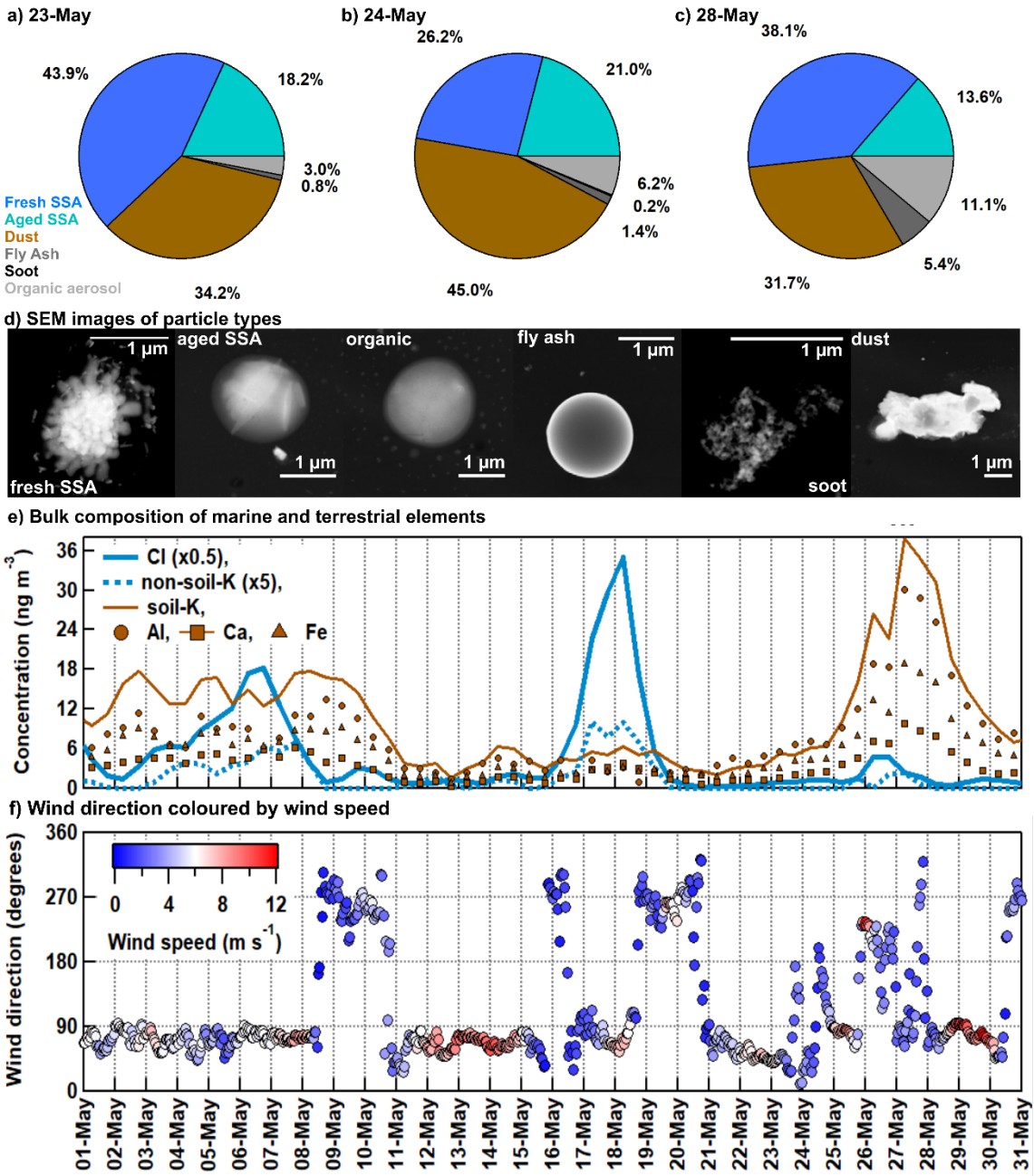

**Figure 7: Summary of results from single-particle and bulk compositional measurements during the late May time period. Relative number fractions of CCSEM-EDX single-particle types for samples (> 1.15 μm) for a) 23 May (1156 particles), a) 24 May (918 particles), and c) 28 May (723 particles). SSA represents sea spray aerosol. Examples of SEM images of the particle types classified are shown in d), with the length of the bar scaled to 1 μm. S-XRF results for May for elements in the 0.75 to 5.0 μm size range associated with either marine (Cl and non-soil-K) or terrestrial (soil-K, Al, Ca, and Fe) sources are shown in e). Cl and non-soil-K concentrations are provided as ×0.5 and ×5 their concentrations, respectively, to show on the same axis. f) Wind direction is also shown, coloured by wind speed. For both CCSEM-EDX and S-XRF classifications, shades of blue and brown represent marine and terrestrial sources, respectively.**

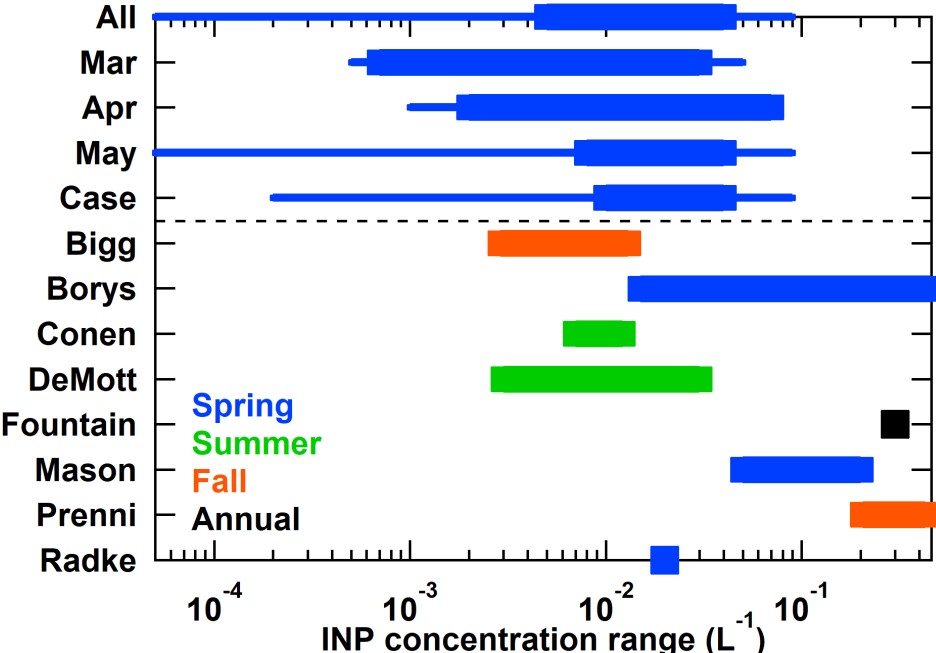

**Figure 8: Summary of ranges of atmospheric immersion mode INP concentrations from the current study and those previously reported in the Arctic. Study details can be found in Table 2. For the current study, "all" and "case" correspond to samples from the entire INPOP study and to the 22 – 29 May time period, respectively. Additionally, the bars and whiskers represent the range between the minimum and maximum INP concentrations at the four temperatures in Table 2 per study. Measurements are coloured to the season in which they are predominantly collected during. The dashed line separates results from the current study from those presented in previous works.**