# Peer review of "Marine and terrestrial influences on ice nucleating particles during continuous springtime measurements in an Arctic oilfield location"

_Atmospheric Chemistry and Physics, 2018_

## Referee Comment (RC1) · Anonymous Referee #1 · 20 Jul 2018

Review of "Marine and terrestrial influences on ice nucleating particles during continuous springtime measurements in an Arctic oilfield location" by Creamean et al. The authors report measurements of ice nucleating particles (INPs) over the period of 3 months in the Arctic during spring. Data from 17 days are presented. Since there are few measurements of INPs in this region and INPs are important for predicting climate, the measurements are certainly valuable. However, most of the conclusions reached by the authors are not well supported by the current analysis. The authors have a nice data set, but a more rigorous analysis is needed to support the conclusions in this manuscript. Specific comments are included below.

[Figure]

Comments:

Abstract. The authors state that the concentrations of coarse mode INP at -15 C were low during the first part of the campaign but then increased by nearly 2 orders during late May. To better illustrate these results, please use a log scale in Fig. 4 to represent the INP concentrations. Currently, the changes in INP concentrations at -15 C are not clear in Fig. 4 due to the use of a linear scale to represent all freezing temperatures.

Abstract. The authors state that the higher concentrations were attributed to air masses originating from over sea ice leads and tundra surfaces. This conclusion was mainly based on back trajectories and information on snow and ice coverage in the Arctic. Although reasonable, the analysis was not very rigorous. The authors did not rule out anthropogenic pollution from local sources. This should be done in a rigorous and quantitative manner if the authors want to claim the INPs are from natural sources.

Section 1.2 (Sample collection). For INP analysis, particles were deposited on Vaseline-coated Mylar. Could the Vaseline influence the INPs and shut off freezing of some of the INPs? Were all INPs extracted from the Vaseline coating? Since this is a new approach and Vaseline could cause artifacts, the influence of Vaseline on INP concentrations should be discussed.

Page 8, lines 19-21. The authors refer to previous studies to suggest that they had relatively clean conditions around late May. But the references are from different years. Hence, these references may not precisely apply to the current study. The authors need to prove that late May was associated with relatively clean conditions. This is not obvious from Figure 2. This is especially important if the authors want to claim the INPs are from natural sources.

Page 9, line 11. The larger spread in INP concentrations could also be due to just a larger variation in source strength.

Page 9, line 28-29. "particles that are theoretically thought to be too small to serve as

efficient INPs." I do not think these references presented a theory that suggested that small particles are not efficient INPs. There are also several laboratory studies that show INPs can have sizes similar to the ranges isolated by stages C and D.

Page 10, line 6-8. The authors suggest that the back trajectories (Figure 6) illustrate that the transport was slow and remained near the surface. However, there is no information on trajectory height or time in Figure 6. I suggest that Figure 6 be modified to include height information and time information to support the author's claims. Also, the trajectories go outside the plotting area in Figure 6. I suggest that the plotting area be increased so the full trajectories can be observed.

Page 10, line 13-15. "Air reaching Oliktok point originated from over a large area of open leads within 30 km off the coast of Utqiagvik (Figure 6d)." This point is not clear from Figure 6d. Where are the open leads in Figure 6d?

Page 10, line 15-16. "sources were generally regional in nature. . ." This statement is not well supported. As the authors point out, mineral dust can be transported to the Arctic from long distances [Stone et al., 2007]. Perhaps the INPs measured by the authors were mineral dust containing biological material transported from long distances? I do not think this was ruled out. If the authors want to rule out long-range transport than more information and discussion is required.

Page 12, lines 1-5. "We present the first INP measurements in an Arctic oilfield location and demonstrate how local and regional transport from marine and terrestrial sources to an industrial region can introduce high concentrations of coarse, warm temperature INPs that are possibly of biological origin." The authors have not ruled out adequately local industrial emissions as the source of INPs. If they want to claim that the sources were local marine and terrestrial, more analysis is required to rule out local industrial emissions. For example, does the INP concentrations correlate with tracers of industrial emissions?

Figure 4. As mentioned above, I suggest using a log scale to illustrate the INP concentrations. Currently it is very difficult to follow the trend at -10 C and -15 C due to the linear scale. Also, what is the onset 99th percentile, how was this calculated, and why is it different from the measured onset temperature? Is the measured onset temperature or the onset 99th percentile more relevant? Also, Figure 4 doesn't include error bars. Error bars should be included, otherwise it is hard to tell if the trends are statistically significant.

Figure 27, a, b, c. Why only include SEM-EDX results from the end of May? Results from the first part of May would be very useful to interpret the INP data.

Figure 7, panel e. I suggest a separate panel for the wind speed and direction. Currently the panel is congested.

Figure caption 8. "Additionally, the bars and whiskers represent INP concentrations at temperatures in Table 2 and at all temperatures, respectively." What temperatures from Table 2 are plotted? What was plotted in the case of "none" in Table 2? Please include this information in the figure caption for clarity.

---

## Referee Comment (RC2) · Anonymous Referee #2 · 31 Jul 2018

General Comments:

"Marine and Terrestrial influences on ice nucleating particles during continuous springtime measurements in an Arctic oilfield location" by Creamean et al. describes results from a 3-month field campaign in Oliktok Point, Alaska in 2017. The field campaign included detailed measurements of in situ aerosol size distribution and number and offline measurements of aerosol composition and ice nucleating particles (INPs). Utilizing size-resolved aerosol impactors, the authors determined the ice nucleation ability of a range of particles sizes. Further, back trajectory modeling and sea ice and snow cover data were used to investigate the influence of various sources on the measured

INP concentrations. The authors provide some evidence that changes in sea ice and snow cover may influence INP number concentrations at the measurements site, but lack an explanation of the mechanism that triggers emissions of INPs due to changes in sea ice and snow cover or suggestions on how to explore this in future work. While it is stated that the data demonstrate how "efficient, natural INPs are likely important in such a relatively polluted Arctic location", the supporting evidence for this is not clearly presented and it is not obvious how this study differs from other coastal studies that were found to be influenced by non-marine aerosol sources. Additional analysis and/or details would be beneficial for supporting the conclusions of the paper. Nevertheless, these data are a certainly a substantial contribution to the field given the extreme lack of INP observations in the Arctic and a recent surge of interest in advancing scientific understanding of aerosol cloud interactions in polar regions.

Specific Comments:

Abstract:

P1 – L20 – "radiative properties" were not included in the analysis and discussion in the paper.

P1 - L21: What is meant by "efficient" INPs? Do you mean the most efficient based on the nucleation temperature (i.e., coarse mode aerosol froze at a warmer temperature than submicron aerosol samples)? Or do you mean most efficient described by ice nucleation site density (INPs normalized by surface area) or ice nucleation efficiency (INPs normalized by total number of particles)? Or simply that the highest number concentrations of INPs were observed in the coarse mode ?

P1 - L26: Please specify that these data are representative of springtime INP number concentrations at this location (rather than year-round values for the Arctic region). Also, the INP analysis was only performed on 16 days of the 3-month period. This should also be clear.

Introduction:

P2 - L9: "Immersion freezing is the most relevant..." - Please provide a reference for this.

P3 – L11 – Can you elaborate on the terrestrial sources that impacted these other coastal studies? How will this study and approach uniquely address this difficult task of elucidating local terrestrial sources (natural and pollution) from pristine marine sources?

P3 - L27 – What is mean by "natural"?

P3 – L28 – I think the introduction is very well written. The overview of the different types of INPs is good, but I think some background information on the aerosol composition and sources of the Arctic Region is also needed. In particular, what are the potential sources of aerosol (i.e., pollution, transported dust, marine organic aerosol, etc.) and what seasonal and conditions are those aerosol sources present? The authors primarily focus on biological particles, but this is not the only aerosol type in the Arctic.

Methods:

P4 - L13 – Were these collections made at ambient relative humidity? If so, please discuss how this may affect the cut size diameter of each stage.

P5 – L21 – How was the focus period selected? What is mean by "interesting aerosol events"?

P6 - L10 – How were blanks collected? Were multiple blanks collected throughout the study (i.e., at the beginning, during and end?). Only one shown in Figure 3.

Throughout – the section numbers are inconsistent with the rest of the Methods section.

Results and discussion:

P8-L10 – can you say anything about the size distributions of particles during these different atmospheric conditions? E.g., the CPCf/CPCu ratio, UHSAS size distribution, etc?

P8 – L11 – "general relatively high" – relative to what?

P8 – L20 – "resulted in relatively 'cleaner' conditions" – What is implied by the quotations? Should this simply state that the changes in transport and increased precipitation resulted in lower particle concentrations?

P8 – L23 – The predominate wind direction during April and May looks more easterly (mostly red). Perhaps a wind rose plot would be helpful for this discussion?

P8 – L25 – If the goal of this study is the examine the role of pollution versus natural aerosol on the INP populations at this site (I think this is correct, though it is not entirely clear), a section is needed that describes the potential influence of natural vs. pollution particles and how you differentiated the difference particle classes. This of course also requires the aerosol composition during the campaign to be summarized earlier. I suggest that the Results and Discussion section be reorganized to first talk about the aerosol composition and influences of natural and anthropogenic aerosol, followed by a discussion on the INP populations measured at the site with a specific section describing the results that support the statement that was in the abstract: "...demonstrate strong influences from natural sources despite the relatively high pollution levels in this Arctic environment".

Fig 2 – Adding some indicator for days of this campaign that were analyzed with the DFCP will help the reader follow along.

Fig 3 – Are these blank-corrected spectra? If so, it might be better to show the blank in a supplemental figure. If not, the blank spectra should be shown on all four panels.

P9 – L30 – The delta T parameter is presented oddly. What is the physical meaning of this parameter? The delta T here is limited by the temperature in which the DFCP

saturates (i.e., all droplets freeze), not the "range of freezing temperatures". Is the goal to define a parameter that describe the presence of the "hump" of INPs that are active at warmer temperatures? While the delta T parameter will be lower for spectra that have a "hump" of INPs at warmer temperatures and higher for spectra those do not, the delta T parameter could also be lower for an INP spectra with a steep slope compared to a spectra with lower slope. If the authors want to describe the presence of significant differences in the number of INPs active at warmer temperatures, a better variable to use may be the temperature in which 50% of the wells were frozen. Or, perhaps the authors can clarify the meaning of this parameter.

Fig 4. Are there uncertainty bars for the INP number concentrations?

P10 – L7 – Please provide trajectory heights in Fig 6, as you refer to the trajectory height in the text and this is one of the main pieces of evidence provided for a connection between the sea ice leads and the observed aerosol.

P10 – L9 – Are these observations of leads and polynyas from satellite, an aircraft, or published? Since the importance of the observed leads are a critical point to your conclusions, these should be provided in some capacity?

P10 – L12 – Can you provide more information about the gravitational settling? I think particles in the largest stage could survive such a transit, but this could be calculated.

P10 – L19 – on May 29, it looks like there is a portion of the back trajectory (72-73 N and 137-133 W) with lower sea ice percentage compared to any portion of the May 22 source region. How are these two regions distinct? What is considered a significant amount of time to spend over an open lead?

P9-L20 – What is the hypothesized mechanism/source of INPs from open Arctic leads? Is organic marine aerosol the suspect? Are there previous studies to suggest that unique aerosol types may be emitted from Arctic leads? How does wind speed play a role? Were Chl a concentrations available for this region?

Summary:

P12 – L9 – "These higher concentrations are attributed to air masses originating from over sea ice leads and tundra surfaces" – Can the authors elaborate on what these particles are exactly? Or provide a hypothesis of what these may be? The single particle and bulk composition measurements suggests significant influence from mineral dusts, but what would be the mechanism for these particles entering the atmosphere via sea ice leads? Particularly for those that were measured in air masses originating from these open Arctic leads?

Can the authors elaborate on future needs for understanding more about these significant increases in INPs? How can the scientific understanding of Arctic INP population variability advance? More measurements? Different measurements?

---

## Author Comment (AC1) · 10 Oct 2018

*We would like to thank the reviewer for his/her insightful feedback regarding our manuscript. We have revised based on his/her commentary and believe the manuscript is much stronger as a result.*

Review of "Marine and terrestrial influences on ice nucleating particles during continuous springtime measurements in an Arctic oilfield location" by Creamean et al. The authors report measurements of ice nucleating particles (INPs) over the period of 3 months in the Arctic during spring. Data from 17 days are presented. Since there are few measurements of INPs in this region and INPs are important for predicting climate, the measurements are certainly valuable. However, most of the conclusions reached by the authors are not well supported by the current analysis. The authors have a nice data set, but a more rigorous analysis is needed to support the conclusions in this manuscript. Specific comments are included below.

Comments:

Abstract. The authors state that the concentrations of coarse mode INP at -15 C were low during the first part of the campaign but then increased by nearly 2 orders during late May. To better illustrate these results, please use a log scale in Fig. 4 to represent the INP concentrations. Currently, the changes in INP concentrations at -15 C are not clear in Fig. 4 due to the use of a linear scale to represent all freezing temperatures.

*Thank you for the suggestion. We have changed Figure 4 so that log scale is shown.*

Abstract. The authors state that the higher concentrations were attributed to air masses originating from over sea ice leads and tundra surfaces. This conclusion was mainly based on back trajectories and information on snow and ice coverage in the Arctic. Although reasonable, the analysis was not very rigorous. The authors did not rule out anthropogenic pollution from local sources. This should be done in a rigorous and quantitative manner if the authors want to claim the INPs are from natural sources.

*The air mass trajectory analysis was used for context for the chemical analysis conducted, both of single particle and bulk compositional information, in addition to other supporting information. First and foremost, we know that the aerosol composition in general in the size ranges relevant to the INP measurements were predominantly sea spray aerosol and dust based on the chemical analyses. Very little influences from soot or fly ash (i.e., local industrial pollution) were observed (4% and 16% of the particles that were > 1.15 µm on 23 May and 28 May, respectively; Figure 7). Second, based on size alone, we would not expect pollution sourced from Prudhoe Bay (which a majority by number are sub-100 nm; Creamean et al. (2018a); Maahn et al (2017)) to overlap with the sizes of the INPs observed (i.e., > 2.96 µm) at Oliktok Point. Third, INPs measured at the warmer end of the temperatures we focus on during our case study are likely biological or dust in origin (e.g., Kanji et al. (2017), Murray et al. (2012)). Fly ash and soot generally form ice at much colder temperatures. We have revised Figure 7 to show wind direction and speed separately, and now discuss this in more detail at the end of section 3.3. We note that although winds were easterly on 28 and 29 May, wind prior to those days were variable and can help explain the aerosol sources during our higher marine and terrestrial INP concentration periods. Thus, based on the combination of freezing temperatures, size, single-particle composition, bulk composition, local meteorology, and air mass transport, we demonstrate that there was indeed little influence from local anthropogenic pollution. We added several sentences discussing these points at the end of section 3.3—that it is possible but unlikely that local pollution largely influenced the INP concentrations during late May.*

Section 1.2 (Sample collection). For INP analysis, particles were deposited on Vaseline-coated Mylar. Could the Vaseline influence the INPs and shut off freezing of some of the INPs? Were all INPs extracted from the Vaseline coating? Since this is a new approach and Vaseline could cause artifacts, the influence of Vaseline on INP concentrations should be discussed.

*The fact that Vaseline is used in preparation of the copper plates and contributes very little artifacts to the INPs during testing indicates it likely does not contribute significant artifacts to the samples containing aerosols impacted on Vaseline. The use of the Vaseline coating has already been tested and discussed in Creamean et al. (2018b) as we note in the methods section. Additionally, Tobo et al. (2016) presented drop freezing results using Vaseline with little to no contaminant influences from the Vaseline itself, and observed blank water samples freezing starting at –30 ℃. However, we cannot determine if all the INPs were extracted with certainty. For clarity, we have added, "It is possible not all particles were removed during the extraction process; however, previous control testing indicates sufficient aerosol loading is resuspended (Creamean et al., 2018b)." to section 1.2.*

Page 8, lines 19-21. The authors refer to previous studies to suggest that they had relatively clean conditions around late May. But the references are from different years. Hence, these references may not precisely apply to the current study. The authors need to prove that late May was associated with relatively clean conditions. This is not obvious from Figure 2. This is especially important if the authors want to claim the INPs are from natural sources.

*The purpose of providing the average values of particle number concentrations in the text, specifically from the CPCs which are indicative of Prudhoe Bay pollution (Creamean et al. (2018a); Maahn et al. (2017)) was to demonstrate how May was relatively clean compared to March and April. To make this clearer, we have revised Figure 2 to show hourly averages and log scale for the particles concentrations to demonstrate the differences between Mar and Apr versus May. We also changed the average values provided in the text to those calculated from hourly concentrations to reflect the values in the figure. One interesting feature that is more prominent is the high variability with very low to concentrations comparable to the previous two months in May. We now discuss this in the text, in addition to highlight such variability as part of the "interesting aerosol events". Please note that we have also removed the wind parameters panel as that was redundant to Figure 7 and we do not discuss the local winds until later when describing the particle compositions.*

Page 9, line 11. The larger spread in INP concentrations could also be due to just a larger variation in source strength.

*Thank you for pointing this out. We have added this possibility.*

Page 9, line 28-29. "particles that are theoretically thought to be too small to serve as efficient INPs." I do not think these references presented a theory that suggested that small particles are not efficient INPs. There are also several laboratory studies that show INPs can have sizes similar to the ranges isolated by stages C and D.

*We have revised the wording to instead state that particles of these sizes are relatively less efficient INPs as compared to their larger counterparts.*

Page 10, line 6-8. The authors suggest that the back trajectories (Figure 6) illustrate that the transport was slow and remained near the surface. However, there is no information on trajectory height or time in Figure 6. I suggest that Figure 6 be modified to include height information and time information to support the author's claims. Also, the trajectories go outside the plotting area in Figure 6. I suggest that the plotting area be increased so the full trajectories can be observed.

*We have revised Figure 6 to include height and time information. We also increased plot size for the full trajectories and included zoomed in panels to show more local scale transport.*

Page 10, line 13-15. "Air reaching Oliktok point originated from over a large area of open leads within 30 km off the coast of Utqiagvik (Figure 6d)." This point is not clear from Figure 6d. Where are the open leads in Figure 6d?

*We have revised Figure 6 (see response to previous comment) to include an inset of the areas of open water within 30 km and describe this feature more in section 3.3. Open leads are evidenced by the regions that are < 100% sea ice percentage, indicating the presence of open water in the 4 km grid cells. However, we realize that the term "leads" may not accurately represent the area of < 100% sea ice concentrations (i.e. light blue colors directly north of Oliktok Point). The width of leads varies from a couple of meters to over a kilometer, thus, they are difficult to resolve in the 4-km sea ice data we use. It is possible the open water is simply small or larger ice floes that have broken off the pack ice near the ice edge. Thus, we have changed this to "marginal ice zone" (MIZ) as that is a more accurate description of the transition between open water and sea ice during the melt season, but that leads may be a feature within this zone.*

Page 10, line 15-16. "sources were generally regional in nature. . ." This statement is not well supported. As the authors point out, mineral dust can be transported to the Arctic from long distances [Stone et al., 2007]. Perhaps the INPs measured by the authors were mineral dust containing biological material transported from long distances? I do not think this was ruled out. If the authors want to rule out long-range transport than more information and discussion is required.

*It is possible that the INPs we observed could have originated from mineral dust from distant sources. However, we provide several reasons as to why the INPs were likely from more local or regional sources. Recent work by Kylling et al. (2018) demonstrates surface dust impacts from several midlatitude sources to the entire Arctic is predominant during May. Yet, they also show that the major source of dust at the surface is from North American north of 60 °N, indicating regional source influences. Specific to our region, supermicron mineral dust has been shown to be transported during Arctic Haze as demonstrated by Quinn et al. (2002) at Barrow, but at very low mass concentrations relative to other aerosol species (i.e., < 0.005 µg m$^{-3}$ based on nss-Ca$^{2+}$). Additionally, given the size in which we observed the INPs (> 2.96 µm), it is unlikely such large particles were transported from very distant sources, especially considering the air mass transport pathways were near the surface in the boundary layer for several days prior to arrival at the site (evidenced by the revised Figure 6). We have added a few sentences ruling this possible source out in section 3.3.*

*Kylling, A., Zwaaftink, C. D. G., and Stohl, A.: Mineral Dust Instantaneous Radiative Forcing in the Arctic, Geophys Res Lett, 45, 4290-4298, 10.1029/2018gl077346, 2018.*
*Quinn, P. K., Miller, T. L., Bates, T. S., Ogren, J. A., Andrews, E., and Shaw, G. E.: A 3-year record of simultaneously measured aerosol chemical and optical properties at Barrow, Alaska, J Geophys Res-Atmos, 107, Artn 4130 10.1029/2001jd001248, 2002.*

Page 12, lines 1-5. "We present the first INP measurements in an Arctic oilfield location and demonstrate how local and regional transport from marine and terrestrial sources to an industrial region can introduce high concentrations of coarse, warm temperature INPs that are possibly of biological origin." The authors have not ruled out adequately local industrial emissions as the source of INPs. If they want to claim that the sources were local marine and terrestrial, more analysis is required to rule out local industrial emissions. For example, does the INP concentrations correlate with tracers of industrial emissions?

*Please see response to the second comment.*

Figure 4. As mentioned above, I suggest using a log scale to illustrate the INP concentrations. Currently it is very difficult to follow the trend at -10 C and -15 C due to the linear scale. Also, what is the onset 99th percentile, how was this calculated, and why is it different from the measured onset temperature? Is the

measured onset temperature or the onset 99th percentile more relevant? Also, Figure 4 doesn't include error bars. Error bars should be included, otherwise it is hard to tell if the trends are statistically significant.

*Please see response to the first comment. The 99th percentile was removed from the figure to avoid confusion, especially since we do not discuss this value in the results and discussion. We have also added error bars to the INP concentrations and onset temperatures.*

Figure 27, a, b, c. Why only include SEM-EDX results from the end of May? Results from the first part of May would be very useful to interpret the INP data.

*We agree that results from early May would be useful; however, given limited resources and samples, we were not able to analyze any more samples other than those presented. The bulk composition was measured during this time period, and we do compare to the case study time period.*

Figure 7, panel e. I suggest a separate panel for the wind speed and direction. Currently the panel is congested.

*Done.*

Figure caption 8. "Additionally, the bars and whiskers represent INP concentrations at temperatures in Table 2 and at all temperatures, respectively." What temperatures from Table 2 are plotted? What was plotted in the case of "none" in Table 2? Please include this information in the figure caption for clarity.

*We have clarified that the bars show the ranges of INP concentrations (i.e., minimum and maximum at the four temperatures shown in Table 2) in the caption.*

---

## Author Comment (AC2) · 10 Oct 2018

*We would like to thank the reviewer for his/her insightful feedback regarding our manuscript. We have revised based on his/her commentary and believe the manuscript is much stronger as a result.*

General Comments:

"Marine and Terrestrial influences on ice nucleating particles during continuous springtime measurements in an Arctic oilfield location" by Creamean et al. describes results from a 3-month field campaign in Oliktok Point, Alaska in 2017. The field campaign included detailed measurements of in situ aerosol size distribution and number and offline measurements of aerosol composition and ice nucleating particles (INPs). Utilizing size-resolved aerosol impactors, the authors determined the ice nucleation ability of a range of particles sizes. Further, back trajectory modeling and sea ice and snow cover data were used to investigate the influence of various sources on the measured INP concentrations. The authors provide some evidence that changes in sea ice and snow cover may influence INP number concentrations at the measurements site, but lack an explanation of the mechanism that triggers emissions of INPs due to changes in sea ice and snow cover or suggestions on how to explore this in future work. While it is stated that the data demonstrate how "efficient, natural INPs are likely important in such a relatively polluted Arctic location", the supporting evidence for this is not clearly presented and it is not obvious how this study differs from other coastal studies that were found to be influenced by non-marine aerosol sources. Additional analysis and/or details would be beneficial for supporting the conclusions of the paper. Nevertheless, these data are a certainly a substantial contribution to the field given the extreme lack of INP observations in the Arctic and a recent surge of interest in advancing scientific understanding of aerosol cloud interactions in polar regions.

Specific Comments:

Abstract:

P1 – L20 – "radiative properties" were not included in the analysis and discussion in the paper.

*True. We have removed "radiative properties" in addition to "chemistry", since we do not present routine chemistry (i.e., the chemistry presented in the manuscript was measured during the intensive period only).*

P1 - L21: What is meant by "efficient" INPs? Do you mean the most efficient based on the nucleation temperature (i.e., coarse mode aerosol froze at a warmer temperature than submicron aerosol samples)? Or do you mean most efficient described by ice nucleation site density (INPs normalized by surface area) or ice nucleation efficiency (INPs normalized by total number of particles)? Or simply that the highest number concentrations of INPs were observed in the coarse mode?

*We meant based on freezing temperature and have reworded this line to reflect that.*

P1 - L26: Please specify that these data are representative of springtime INP number concentrations at this location (rather than year-round values for the Arctic region). Also, the INP analysis was only performed on 16 days of the 3-month period. This should also be clear.

*We clarified that this was springtime. We chose not to add the number of analysis days to the abstract, as we wanted to focus on the results themselves and particularly on the May case study period. The trends we observed are justified by our statements. Details on how many samples were analyzed are clarified throughout the manuscript.*

Introduction:

P2 - L9: "Immersion freezing is the most relevant. . . " - Please provide a reference for this.

*We have added Hande and Hoose (2017).*

*Hande, L. B., and Hoose, C.: Partitioning the primary ice formation modes in large eddy simulations of mixed-phase clouds, Atmos Chem Phys, 17, 14105-14118, 10.5194/acp-17-14105-2017, 2017.*

P3 – L11 – Can you elaborate on the terrestrial sources that impacted these other coastal studies? How will this study and approach uniquely address this difficult task of elucidating local terrestrial sources (natural and pollution) from pristine marine sources?

*We have added more background regarding the sources impacting the previous coastal studies mentioned in the fourth paragraph of the introduction. Our unique angle is that we use a comprehensive combination of size-resolved INP measurements, single-particle chemistry, bulk chemistry, local meteorology, regional scale transport, and sea ice and land cover conditions to assess INP sources. We now state this at the end of the introduction.*

P3 - L27 – What is mean by "natural"?

*We clarified that we meant "naturally-sourced".*

P3 – L28 – I think the introduction is very well written. The overview of the different types of INPs is good, but I think some background information on the aerosol composition and sources of the Arctic Region is also needed. In particular, what are the potential sources of aerosol (i.e., pollution, transported dust, marine organic aerosol, etc.) and what seasonal and conditions are those aerosol sources present? The authors primarily focus on biological particles, but this is not the only aerosol type in the Arctic.

*Thank you for the comment. Although general Arctic aerosol composition and sources are certainly important to discuss, a comprehensive review of Arctic aerosol is outside the scope of this manuscript. Our study location is unique in that it is an Arctic oilfield. Additionally, oilfield locations are subject to very different aerosol sources as compared to typical Arctic background locations. Unfortunately, very few aerosol studies have been conducted in oilfield locations, which would be parallel to our measurements. We have elaborated on the local sources of aerosol observed during the summer from our previous studies (Creamean et al. (2018); Maahn et al. (2017)), in addition to a couple other recent studies by Gunsch et al. (2017) and Kirpes et al. (2018) at the end of the introduction to provide more context for what has been observed in the Alaskan Arctic for oilfield aerosol influences.*

Methods:

P4 - L13 – Were these collections made at ambient relative humidity? If so, please discuss how this may affect the cut size diameter of each stage.

*These were collected at ambient RH, and as a result we have added a sentence stating how RH may affect particle size by making them larger. However, the purpose of collecting at ambient RH is to mimic how the aerosols themselves would nucleate ice in the environmental conditions they existed in.*

P5 – L21 – How was the focus period selected? What is mean by "interesting aerosol events"?

*This period was chosen based on the shift in air mass sources and large variability particle concentrations and discussed in the results and discussion. We have clarified this in the text as well and revised Figure 2 to show hourly averaged particle concentrations in which the variability of the aerosol is more evident.*

P6 - L10 – How were blanks collected? Were multiple blanks collected throughout the study (i.e., at the beginning, during and end?). Only one shown in Figure 3.

*Only one blank was collected and tested during the analysis phase. However, previous control study testing has demonstrated the reliability of the pre-treated and prepared PFA (Creamean et al. (2018b)).*

Throughout – the section numbers are inconsistent with the rest of the Methods section.

*Fixed.*

Results and discussion:

P8-L10 – can you say anything about the size distributions of particles during these different atmospheric conditions? E.g., the CPCf/CPCu ratio, UHSAS size distribution, etc?

*We have added the hourly-averaged mean particle diameters from the UHSAS to Figure 2b and discuss in section 3.1. Unfortunately, the size ranges of the CPCs and UHSAS are well below the stage A sizes, so we cannot disseminate the mean size results much beyond describing initial conditions (i.e., we cannot use these results to support the 2.96 – > 12 μm INPs). Sizing measurements at sizes relevant to stage A INPs were not available.*

P8 – L11 – "general relatively high" – relative to what?

*To other locations on the North Slope, which we now have clarified here.*

P8 – L20 – "resulted in relatively 'cleaner' conditions" – What is implied by the quotations? Should this simply state that the changes in transport and increased precipitation resulted in lower particle concentrations?

*We removed "relatively 'cleaner' conditions" and revised to the suggested change.*

P8 – L23 – The predominate wind direction during April and May looks more easterly (mostly red). Perhaps a wind rose plot would be helpful for this discussion?

*We removed this sentence since the wind data are more relevant in the following sections when discussing the chemistry (we removed the wind panel from the figure as well). We have also revised Figure 7 to include a panel of just wind direction and speed. We now discuss the winds more closely at the end of section 3.3.*

P8 – L25 – If the goal of this study is the examine the role of pollution versus natural aerosol on the INP populations at this site (I think this is correct, though it is not entirely clear), a section is needed that describes the potential influence of natural vs. pollution particles and how you differentiated the difference particle classes. This of course also requires the aerosol composition during the campaign to be summarized earlier. I suggest that the Results and Discussion section be reorganized to first talk about the aerosol composition and influences of natural and anthropogenic aerosol, followed by a discussion on the INP populations measured at the site with a specific section describing the results that support the statement that was in the abstract: ". . .demonstrate strong influences from natural sources despite the relatively high pollution levels in this Arctic environment".

*We disagree that our ordering does not support the natural versus pollution sources of INPs and that restructuring the results and discussion would support this. The idea behind the ordering was to first discuss*

*the conditions observed generally during the study, then show the large shift in INPs, followed by the evidence to support why this shift happened and what the likely sources of the INPs were.*

*We also believe we have enough supporting evidence for demonstrating the INPs were likely from local and regional natural sources during our late May case study. The air mass trajectory analysis was used for context for the chemical analysis conducted, both of single particle and bulk compositional information, in addition to other supporting information. First and foremost, we know that the aerosol composition in general in the size ranges relevant to the INP measurements were predominantly sea spray aerosol and dust based on the chemical analyses. Very little influences from soot or fly ash (i.e., local industrial pollution) were observed (4% and 16% of the particles that were > 1.15 μm on 23 May and 28 May, respectively; Figure 7). Second, based on size alone, we would not expect pollution sourced from Prudhoe Bay (which a majority by number are sub-100 nm; Creamean et al. (2018a); Maahn et al (2017)) to overlap with the sizes of the INPs observed (i.e., > 2.96 μm) at Oliktok Point. Third, INPs measured at the warmer end of the temperatures we focus on during our case study are likely biological or dust in origin (e.g., Kanji et al. (2017), Murray et al. (2012)). Fly ash and soot generally form ice at much colder temperatures. We have revised Figure 7 to show wind direction and speed separately, and now discuss this in more detail at the end of section 3.3. We note that although winds were easterly on 28 and 29 May, wind prior to those days were variable and can help explain the aerosol sources during our higher marine and terrestrial INP concentration periods. Thus, based on the combination of freezing temperatures, size, single-particle composition, bulk composition, local meteorology, and air mass transport, we demonstrate that there was indeed little influence from local anthropogenic pollution. We added several sentences discussing these points at the end of section 3.3—that it is possible but unlikely that local pollution largely influenced the INP concentrations during late May.*

Fig 2 – Adding some indicator for days of this campaign that were analyzed with the DFCP will help the reader follow along.

*Done.*

Fig 3 – Are these blank-corrected spectra? If so, it might be better to show the blank in a supplemental figure. If not, the blank spectra should be shown on all four panels.

*They are not blank corrected. We have added the blanks to each of the four panels.*

P9 – L30 – The delta T parameter is presented oddly. What is the physical meaning of this parameter? The delta T here is limited by the temperature in which the DFCP saturates (i.e., all droplets freeze), not the "range of freezing temperatures". Is the goal to define a parameter that describe the presence of the "hump" of INPs that are active at warmer temperatures? While the delta T parameter will be lower for spectra that have a "hump" of INPs at warmer temperatures and higher for spectra those do not, the delta T parameter could also be lower for an INP spectra with a steep slope compared to a spectra with lower slope. If the authors want to describe the presence of significant differences in the number of INPs active at warmer temperatures, a better variable to use may be the temperature in which 50% of the wells were frozen. Or, perhaps the authors can clarify the meaning of this parameter. Fig 4. Are there uncertainty bars for the INP number concentrations?

*After consideration, we decided to remove ΔT from the manuscript and Figure 5 since we did not discuss it in detail and it did not add additional useful information to the main conclusions. The information from the INP concentrations and onset freezing temperatures sufficiently supports our main conclusions.*

P10 – L7 – Please provide trajectory heights in Fig 6, as you refer to the trajectory height in the text and this is one of the main pieces of evidence provided for a connection between the sea ice leads and the observed aerosol.

*Done. We have substantially revised Figure 6 to include such information and to support our conclusions in the text.*

P10 – L9 – Are these observations of leads and polynyas from satellite, an aircraft, or published? Since the importance of the observed leads are a critical point to your conclusions, these should be provided in some capacity?

*As we state in the methods, these data are satellite derived. We realize that the term "leads" may not accurately represent the area of < 100% sea ice concentrations (i.e. light blue colors directly north of Oliktok Point). The width of leads varies from a couple of meters to over a kilometer, thus, they are difficult to resolve in the 4-km sea ice data we use. It is possible the open water is simply small or larger ice floes that have broken off the pack ice near the ice edge. Thus, we have changed this to "marginal ice zone" (MIZ) as that is a more accurate description of the transition between open water and sea ice during the melt season, but that leads may be a feature within this zone. We also have defined that the persistent open water regions west of the Canadian Arctic Archipelago and western Alaska are polynyas.*

P10 – L12 – Can you provide more information about the gravitational settling? I think particles in the largest stage could survive such a transit, but this could be calculated.

*The basic terminal settling velocities can be calculated, but this information does not take into account external vertical updrafts or downdrafts and how those features may affect particle lifetime. We did do such calculations and have added discussion on results from such calculations for 16 May and 29 May in the second paragraph in section 3.3, which indicate sources local to Oliktok Point. Additionally, Jaenicke (1980) concludes that particles of these sizes originating within the boundary layer typically reside in the atmosphere for on the order of hours to days (but less than a week). However, for 22 May, air masses did not travel over any substantial open water polynyas rather only over the MIZ north of land, thus, the only possible explanation for the sources on this day is from the open water in this region. Given this information, it is possible these coarse mode INPs could originate from the open water 700 km away, but the more probably scenario is transport from 30 km away. We have also added a couple sentences to elaborate on the gravitational settling based on Jaenicke (1980) and how the distant open water could be a source, but likely not the major source.*

P10 – L19 – on May 29, it looks like there is a portion of the back trajectory (72-73 N and 137-133 W) with lower sea ice percentage compared to any portion of the May 22 source region. How are these two regions distinct? What is considered a significant amount of time to spend over an open lead?

*We have substantially revised Figure 6 to show more detail of the transport pathways and have adjusted the text in this transport paragraph accordingly. Regarding the transport over the lead on 29 May, transport from this region occurred 3 days prior and the air mass traveled very close to the surface, indicating possible gravitational settling during transport. However, INP concentrations were still high relative to days in Mar or Apr, which could be a result of transport over the MIZ. We have now added discussion on this topic to the transport paragraph.*

P9-L20 – What is the hypothesized mechanism/source of INPs from open Arctic leads? Is organic marine aerosol the suspect? Are there previous studies to suggest that unique aerosol types may be emitted from Arctic leads? How does wind speed play a role? Were Chl a concentrations available for this region?

*Recent work by May et al. (2016) has demonstrated that production of sea salt aerosol in the Arctic can occur year-round from leads under elevated wind speeds (i.e., winds speeds > 4 m s$^{-1}$). The main mechanisms behind aerosol production from open ocean surfaces is bubble bursting from wind-induced wave breaking, although this process is far less studied over leads. A recent study by Gabric et al. (2018) nicely describes generation of marine biogenic aerosol (MBA) from sea ice leads and the MIZ. Thus, other primary aerosols may be generated by the same mechanisms that produce sea salt aerosol and MBA. Recent studies by Wilson et al. (2015) and Irish et al. (2017; and references therein) have shown that the Arctic Ocean surface microlayer and bulk seawater can harbor large concentrations of INPs, indicating physical mechanisms that generate aerosol from the surface waters may eject these INPs into the atmosphere. Additionally, several previous high Arctic ice nucleation studies have demonstrated that leads and other open water sources as vital to influencing atmospheric INP concentrations (Bigg, 1994; Bigg and Leck, 2001). Based on a combination of conclusions from this body of previous work, we conclude that INPs from leads or other open water features (i.e., small ice floe regions) are likely produced via waves and/or bubble bursting and particularly under relatively windy conditions and are likely composed of bacteria or fragments of marine organisms.*

*We have added discussion on these previous studies to section 3.3 to support our conclusions. However, because we do not have wind speed measurement over the MIZ region, we cannot quantitatively comment on the role of wind speed over the open water we observed. In situ chlorophyll measurements were not available for this region. We did check chl-a concentrations from MODIS (available at an 8-day time resolution), and although chl-a looks to be elevated in the polynya west of Alaska, cloud cover makes it difficult to discern any sort of temporal trend on the scale of the MIZ north of Oliktok Point:*

[Figure]

*Irish et al. (2017) also evaluated their INP measurements in the context of satellite-based chl-a and were not able to use chl-a concentrations to explain their INP observations.*

*Gabric, A., Matrai, P., Jones, G., and Middleton, J.: The Nexus between Sea Ice and Polar Emissions of Marine Biogenic Aerosols, Bulletin of the American Meteorological Society, 99, 61-82, 10.1175/Bams-D-16-0254.1, 2018.*

*May, N. W., Quinn, P. K., McNamara, S. M., and Pratt, K. A.: Multiyear study of the dependence of sea salt aerosol on wind speed and sea ice conditions in the coastal Arctic, J Geophys Res-Atmos, 121, 9208-9219, 10.1002/2016jd025273, 2016.*

Summary:

P12 – L9 – "These higher concentrations are attributed to air masses originating from over sea ice leads and tundra surfaces" – Can the authors elaborate on what these particles are exactly? Or provide a hypothesis of what these may be? The single particle and bulk composition measurements suggests

significant influence from mineral dusts, but what would be the mechanism for these particles entering the atmosphere via sea ice leads? Particularly for those that were measured in air masses originating from these open Arctic leads? Can the authors elaborate on future needs for understanding more about these significant increases in INPs? How can the scientific understanding of Arctic INP population variability advance? More measurements? Different measurements?

*See response to above. Based on previous work and our chemical measurements, we conclude that the INPs from over the open water and land were primary marine aerosol and dust, respectively. Though each analyzed sample's source influences were characterized based on air mass trajectories, the diversity in chemical composition of the aerosol particles in each sample indicate a variety of sources. The observed dust could be due to local road dust in addition to terrestrial dust sources along the air mass back trajectories. We have now noted this in section 3.3.*

*We already state that additional measurements in seasons other than the spring are needed but have elaborated that such measurements include comprehensive INP concentrations and characterization. A detailed discussion on the future needs is indeed important but outside the scope of this manuscript, and would be better-suited for some sort of Arctic INP review paper.*

---

## Author Response (AR2)

**Reviewer 1**

I thank the referees for carefully addressing my initial comments. The paper is significantly improved. However, I still have comments that need to be addressed before publication in ACP.

Page 6, line 16. The authors state that sufficient aerosol loading is re-suspended during the extraction process. I certainly agree with this, but this does not rule out artifacts. The fraction of material re-suspended may depend on the overall loading, which could lead to artifacts. This should be acknowledged.

*We have acknowledged this possibility on page 6.*

Figure 3. Please label each panel with the correct stage name and size cut. I could not tell what panel corresponds to Stage A – D.

*Thank you for pointing this out. Done.*

Figure 7, Panel e. Based on Figure 7, Panel e, the concentration of mineral dust between May 25 to May 31 is on the order of 60 micrograms/m^3. I believe this is 1-2 orders of magnitude higher than average values typically observed in the arctic during spring-summer. These mineral dust concentrations are also correlated with the high INP concentrations. Based on this observation, I am wondering if mineral dust can explain all the data in Figure 4, panel A. To confirm this (or rule this out) please perform a simple back-of-the-envelope calculation to estimate the INP concentrations from mineral dust at -5, -10, and -15 C. Assume the average diameter of mineral dust is roughly 5 micrometers, the mass of mineral dust is roughly 60 micrograms/m^3, and ns-values for mineral dust (Niemand et al., 2012). Also, please include this type of calculation in your manuscript to support your conclusions that the high temperature INPs are likely biological.

*Thank you for bringing this to our attention, the unit in the figure were wrong and it should be ng m$^{-3}$, which explains the offset be several orders of magnitude. Given we have the correct values now that are comparable to what is typical of the Arctic, we did not conduct the Niemand et al. (2012) calculation. Certainly, it would be interesting to look at our INP spectra relative to the Niemand et al. dust INP and McCluskey et al. (2018) clean marine INP parameterizations; however, one issue is that we do not have surface area measurements for particles > 1 μm at Oliktok Point, so the calculated $n_s$ would not be representative of the DRUM INPs, in which the most efficient ones were > 3 μm.*

*McCluskey, C. S., Ovadnevaite, J., Rinaldi, M., Atkinson, J., Belosi, F., Ceburnis, D., ... DeMott, P. J. (2018). Marine and Terrestrial Organic Ice-Nucleating Particles in Pristine Marine to Continentally Influenced Northeast Atlantic Air Masses. Journal of Geophysical Research: Atmospheres, 123(11), 6196–6212. doi:10.1029/2017jd028033.*

The authors indicate that the mineral dust may be coming from local road dust in addition to terrestrial dust sources. In this case, the dust would not be considered natural, correct? This should be mentioned in the abstract and summary.

*Technically, the dust itself is still natural, but generated by human activity (so not a natural influence). The reason we worded it as such is to differentiate dust and marine sources from pollution such as black carbon. We have revised to state "mineral and marine" instead of "natural".*

Niemand, M., Moehler, O., Vogel, B., Vogel, H., Hoose, C., Connolly, P., Klein, H., Bingemer, H., DeMott, P., Skrotzki, J., and Leisner, T.: A Particle-Surface-Area-Based Parameterization of Immersion

Freezing on Desert Dust Particles, Journal of the Atmospheric Sciences, 69, 3077-3092, 10.1175/jas-d-11-0249.1, 2012.

**Reviewer 2**

Creamean et al., have satisfied most of my concerns and I think the paper should be accepted for publication after addressing the following minor comments.

The revised explanation of the relative humidity and the possible influence on the 50% cutoff of the stages is fine. I agree that this approach makes sense for assessing these particles as they would exist in nature. Can the authors specify if the other instrument were also made at RH (e.g., UHSAS)? Also, this data is compared with the Mason et al. (2015) study, but it's not clear if Mason et al., also were making their measurements at ambient RH or for dry diameters. The concern is for the high RHs, the growth factors (especially in a marine environment) may be quite high. I don't think anything needs to change other than mentioning these two things: 1) was UHSAS was dry/ambient? and 2) was Mason et al., 2015 was dry/ambient?

*The UHSAS was sampled at 40% RH based on the DOE ARM standardized aerosol observing system (AOS) configuration (https://www.arm.gov/publications/tech_reports/handbooks/aos_handbook.pdf). Mason et al. (2016) did not standardize for RH at any of the sites but reported a 69% average RH at six of the seven sites. They state that an RH of 70% or greater significantly reduces particle bounce for impaction methods. RH measured during INPOP was typically > 70% (83±14%), thus limited particle bounce since the DRUM is an impaction-based method like the MOUDI from Mason et al. (2016). We have added the RH for the ARM measurements in the methods and state that dried UHSAS size versus ambient DRUM sizes may vary due to the RH differences. We also added a statement on the RH reported and the bounce affects from Mason et al. (2016) in section 3.2.*

Please use caution in how the DeMott et al. (2010) study is cited with regards to INP size. In P9, L28, DeMott et al. (2010) was used to justify that "modeling studies suggest that INPs can be as small as 500nm" and later that "Stages C and D include particles that are thought to be less efficient INPs relative to particles with larger diameters". However, DeMott et al. (2010) state that 500 nm has been shown as the mode diameter of ice crystal residuals, meaning that 50% of the INP population resides in the particle range less than 0.5 microns. From their paper: "For example, the mode diameter of particles found at the centers of single ice crystals has been reported to be~0.5 μm (26). We have measured a similar mode size for IN collected and analyzed in many of the same studies represented in Fig. 2 (24). "The choice of 0.5-μm diameter as the lower limit for summing number concentrations of "large" particles is a relatively arbitrary one of convenience, selected to limit the influence on derived relationships of high concentrations of non-IN particles in the range 0.1–0.5 μm, while retaining sufficient number concentrations of particles to reference to IN concentrations." Knowledge regarding the size of INPs is limited, which is why the approach used by Creamean et al. is so valuable.

*Noted. We reworded that sentence to, "However, modelling studies suggest the mode of INPs can be as small as 500 nm (DeMott et al., 2010; DeMott et al., 2015; Fridlind et al., 2012), while observational work suggests that nanometre-sized INPs are typically found attached to larger particles in the atmosphere (O'Sullivan et al., 2015)."*

Figure 2: top panel is mislabeled (both right and left are labeled CPCf).

*Thank you for pointing this out. We have fixed the axis labels.*

Figure 3 – All panels are missing the labels that were in the original draft (a-d, with size ranges).

*Thank you for pointing this out. We have re-added the panel labels.*

Figure 5 – Cannot read/follow the colorbar for the top 4 panels.

*We have made the colorbar larger and noted that it is log scale.*

[revised manuscript text omitted]

---

## Author Response (AR3)

Dear Ryan,

We initially did not include the CFDC measurements from Prenni et al. (2009) because they reported total INP concentrations for immersion, deposition, and condensation mode. However, after discussing with Paul, he indicated that *most* of the INPs were dominated by immersion mode. I have now included data Paul provided from M-PACE in Table 2 and Figure 8 with a note indicating the concentrations are mostly from immersion INPs.

Thank you for your suggestion.

Cheers,
Jessie